# CRB3 navigates Rab11 trafficking vesicles to promote γTuRC assembly during ciliogenesis

Bo Wang[1,2†], Zheyong Liang[1,3†], Tan Tan[4], Miao Zhang[1,2], Yina Jiang[5], Yangyang Shang[1,2], Xiaoqian Gao[1,2], Shaoran Song[1,2], Ruiqi Wang[1,2], He Chen[1,2], Jie Liu[1,2], Juan Li[1,2], Yu Ren[6], Peijun Liu[1,2]*

[1]Center for Translational Medicine, the First Affiliated Hospital of Xi'an Jiaotong University, Shaanxi, China; [2]Key Laboratory for Tumor Precision Medicine of Shaanxi Province, the First Affiliated Hospital of Xi'an Jiaotong University, Shaanxi, China; [3]Department of Cardiovascular Surgery, the First Affiliated Hospital of Xi'an Jiaotong University, Shaanxi, China; [4]Center for Precision Medicine, Affiliated to the First People's Hospital of Chenzhou, University of South China, Chenzhou, China; [5]Department of Pathology, the First Affiliated Hospital of Xi'an Jiaotong University, Shaanxi, China; [6]Department of Breast Surgery, the First Affiliated Hospital of Xi'an Jiaotong University, Shaanxi, China

*For correspondence: liupeijun@xjtu.edu.cn

†These authors contributed equally to this work

Competing interest: The authors declare that no competing interests exist.

**Abstract** The primary cilium plays important roles in regulating cell differentiation, signal transduction, and tissue organization. Dysfunction of the primary cilium can lead to ciliopathies and cancer. The formation and organization of the primary cilium are highly associated with cell polarity proteins, such as the apical polarity protein CRB3. However, the molecular mechanisms by which CRB3 regulates ciliogenesis and the location of CRB3 remain unknown. Here, we show that CRB3, as a navigator, regulates vesicle trafficking in γ-tubulin ring complex (γTuRC) assembly during ciliogenesis and cilium-related Hh and Wnt signaling pathways in tumorigenesis. *Crb3* knockout mice display severe defects of the primary cilium in the mammary ductal lumen and renal tubule, while mammary epithelial-specific *Crb3* knockout mice exhibit the promotion of ductal epithelial hyperplasia and tumorigenesis. CRB3 is essential for lumen formation and ciliary assembly in the mammary epithelium. We demonstrate that CRB3 localizes to the basal body and that CRB3 trafficking is mediated by Rab11-positive endosomes. Significantly, CRB3 interacts with Rab11 to navigate GCP6/Rab11 trafficking vesicles to CEP290, resulting in intact γTuRC assembly. In addition, CRB3-depleted cells are unresponsive to the activation of the Hh signaling pathway, while CRB3 regulates the Wnt signaling pathway. Therefore, our studies reveal the molecular mechanisms by which CRB3 recognizes Rab11-positive endosomes to facilitate ciliogenesis and regulates cilium-related signaling pathways in tumorigenesis.

## eLife assessment

This is a **useful** study for scientists interested in cell polarity, epithelial morphogenesis, cancer, and primary cilia. The authors investigate the role of CRB3 in regulating these processes by using a combination of a mammary epithelial cell-specific conditional Crb3 knockout mouse model, and cellular, molecular and biochemical approaches. The results, which are **solid**, supporting and extending previous findings, suggest that CRB3 affects ciliogenesis by a mechanism involving Rab11 and gamma-TuRC.

## Introduction

Epithelial tissues are the most common and widely distributed tissues in the body, and they perform several particular functions. Epithelial tissues are made up of epithelial cells. Establishing apical-basal cell polarity within epithelial cells is a crucial biological process for epithelial homeostasis. Unfortunately, the loss of apical-basal cell polarity is one of the most significant hallmarks of advanced malignant tumors (*Wodarz and Näthke, 2007*), and several reports have shown that the loss of some polarity proteins causes disordered cell polarity within epithelial cells, which cannot tightly control cellular growth and migration, ultimately leading to tumor formation, progression, and metastasis (*McCaffrey and Macara, 2011*; *Etienne-Manneville, 2008*). However, the relationship between polarity proteins and tumorigenesis remains incomplete.

The Crumb complex, located apically, is vital in regulating and maintaining apical-basal cell polarity within epithelial cells. In mammals, three Crumbs orthologs are CRB1, CRB2, and CRB3, and only CRB3 is widely expressed in epithelial cells (*Elsum et al., 2012*). Our previous studies have shown that CRB3 is expressed at low levels in renal and breast cancers, and abnormal CRB3 expression leads to the disrupted organization of MCF10A cells in three-dimensional (3D) cultures (*Li et al., 2017*; *Mao et al., 2017*; *Mao et al., 2015*). Inhibition of CRB3 impairs contact inhibition and leads to migration, invasion, and tumorigenesis of cancer cells (*Mao et al., 2017*; *Karp et al., 2008*). Contact inhibition results in growth arrest and causes epithelial cells to enter the quiescent phase, inducing primary cilium formation.

The primary cilium, an antenna-like microtubule-based organelle, is a sensorial antenna that regulates cell differentiation, proliferation, polarity, and tissue organization in most types of mammalian cells (*Sánchez and Dynlacht, 2016*). Dysfunction of the primary cilium can lead to developmental and degenerative disorders called ciliopathies, such as polycystic kidney disease, nephronophthisis, retinitis pigmentosa, and Meckel syndrome (*Reiter and Leroux, 2017*). Remarkably, many cancers, including melanoma and breast, renal, pancreatic, and prostate cancer, exhibit loss of the primary cilium, most likely during the early stages of tumorigenesis (*Kim et al., 2011*; *Menzl et al., 2014*; *Basten et al., 2013*; *Seeley et al., 2009*; *Hassounah et al., 2013*). The primary cilium is a significant extension on the apical surfaces of epithelial cells and is maintained by polarized vesicular traffic (*Bazellières et al., 2018*). Studies have shown that CRB3b localizes in the primary cilium and that depletion of CRB3 leads to primary cilium loss in Madin–Darby canine kidney (MDCK) cells (*Fan et al., 2007*; *Fan et al., 2004*), indicating that CRB3 is necessary for the initiation of ciliogenesis. However, how CRB3 promotes ciliogenesis on the apical surfaces of epithelial cells remains unknown.

Here, we describe a novel conditional deletion of *Crb3* in mice. Our results demonstrate that Crb3 is required to assemble the primary cilium in the mammary gland, kidney, and mouse embryonic fibroblast (MEF) cells from *Crb3* knockout mice. Moreover, we found that *Crb3* deletion promotes breast cancer progression in vivo. Additionally, we observed that CRB3 knockdown destabilizes the assembly of the γ-tubulin ring complex (γTuRC) during ciliogenesis. Specifically, CRB3 interacts with Rab11 and navigates GCP6/Rab11 trafficking vesicles to the basal body of the primary cilium in mammary epithelial cells. We also identified that CRB3 regulates the ciliary Hedgehog (Hh) and Wnt signaling pathways in breast cancer. Thus, the polarity protein CRB3 is important for the assembly mechanism of the γTuRC in ciliogenesis and the cilium-related signaling pathways in tumorigenesis.

## Results

### *Crb3* deletion mice exhibit smaller size and anophthalmia

Since *Crb3* knockout mice die shortly after birth (*Whiteman et al., 2014*), we generated a novel transgenic mouse model with conditional deletion of *Crb3* using the Cre-loxP system. The loxP sites flanked either side of exon 3 in the *Crb3* gene (*Figure 1—figure supplement 1A*). According to the construction strategy, we are knocking out both isoforms *Crb3a* and *Crb3b* in the mice. The positive embryonic stem (ES) cell clone was confirmed for the wild-type gene (*Crb3*^wt/wt^) and recombinant allele (*Crb3*^wt/fl^) by Southern blotting (*Figure 1—figure supplement 1B*). The genotypes of the wild type, heterozygotes, and homozygotes (*Crb3*^fl/fl^) were detected with specific primers (*Figure 1—figure supplement 1C*). We first intercrossed *Crb3*^fl/fl^ with CMV enhancer/chicken β-actin promoter (CAG)-Cre mice and found that *Crb3*^fl/fl^, *Crb3*^wt/fl^; CAG-Cre and *Crb3*^fl/fl^; CAG-Cre pups were born normally and well developed in size. However, most *Crb3*^fl/fl^; CAG-Cre pups died within a few days

after birth. Few surviving $Crb3^{fl/fl}$; CAG-Cre mice were smaller and showed anophthalmia than littermate $Crb3^{fl/fl}$ mice at 4 weeks old (**Figure 1A and B**). The eye is an organ with perfect apical-basal cell polarity. However, *Crb3* knockout mice showed ocular abnormalities. Together, these results indicate that Crb3 is necessary for eye development.

## Deletion of mammary epithelial-specific *Crb3* causes ductal epithelial hyperplasia and tumorigenesis

Based on our earlier studies, which showed that inhibiting CRB3 impairs the organization of breast epithelium, we set out to directly explore the role of Crb3 in mammary gland development. We first developed epithelial cell-specific deletion of *Crb3* in the mammary gland by intercrossing $Crb3^{fl/fl}$ mice with mouse mammary tumor virus long terminal repeat promoter (MMTV)-Cre mice. We confirmed the expression of Crb3 in the mammary gland using immunoblotting (both isoforms Crb3a and Crb3b), real-time quantitative PCR, and immunohistochemistry, and Crb3 expression was deficient in mammary epithelial cells with Crb3 knockout (**Figure 1—figure supplement 1D–F**).

To assess the function of Crb3 in the development of mammary epithelial cells, we analyzed whole mammary mounts from 8-week-old virgin $Crb3^{fl/fl}$; MMTV-Cre and littermate $Crb3^{fl/fl}$ mice. $Crb3^{fl/fl}$; MMTV-Cre mice displayed significantly increased numbers of terminal end buds (TEBs) and bifurcated TEBs compared to $Crb3^{fl/fl}$ mice (**Figure 1C–E**). Histopathology results revealed that normal ductal epithelial cells were arranged as a monolayer in $Crb3^{fl/fl}$ mice, while *Crb3* knockout led to ductal epithelial hyperplasia with increased ductal thickness in $Crb3^{fl/fl}$; MMTV-Cre mice (**Figure 1F**).

To investigate the role of Crb3 in breast cancer tumorigenesis, we observed mammary glands from 9-week-old virgin polyomavirus middle T antigen (PyMT)-cKO-Crb3 and PyMT-WT mice. The mammary glands of PyMT mice progressed to early carcinomas and lost their normal epithelial architecture (**Ye et al., 2015**). The PyMT mouse model is widely used to study tumor initiation and development, with a histological progression similar to that of human breast cancer. PyMT-WT mice exhibited loss of epithelial features and developed breast cancer, while PyMT-cKO-Crb3 mice developed larger tumors with faster tumor progression and a poorly differentiated phenotype (**Figure 1G**). Thus, these results suggest that Crb3 knockout leads to ductal epithelial hyperplasia and promotes tumorigenesis.

## CRB3 knockdown inhibits lumen formation in acini of mammary epithelial cells

MCF10A cells are spontaneously immortalized human breast epithelial cells with the characteristics of normal breast epithelium (**Soule et al., 1990**). In 3D basement membrane culture conditions, MCF-10A cells form acini similar to mammary epithelial acini. This makes them a valuable system for studying the morphogenesis and oncogenesis of the mammary epithelium (**Halsne et al., 2016**). To confirm the function of CRB3 in mammary epithelial acini formation, we first knocked down the expression of both CRB3a and CRB3b isoforms using lentiviral shRNA in MCF10A cells (**Figure 2—figure supplement 1A and B**). Under the 3D culture system, we found that MCF10A cells infected with a non-targeting shRNA lentivirus formed acini after 6 d (D6) and eventually formed polarized acini with a hollow lumen at D14 (**Figure 2A**). However, after CRB3 knockdown, MCF10A cells formed more and larger aberrant acini without a lumen after D6 (**Figure 2A and B**). Cell proliferation assays of MCF10A cells showed that CRB3 knockdown increased the proliferation rate and accelerated G1 to S phase progression (**Figure 2—figure supplement 1C–E**).

To investigate the reason for aberrant acini without lumens, we further studied apoptosis and mitotic spindle orientation during lumen formation in a 3D culture system. The immunofluorescence (IF) results showed that more internal cleaved-caspase 3-positive acini were observed in the control groups than in the CRB3 knockdown groups at D9 (**Figure 2C and D**). Moreover, CRB3b overexpression significantly promoted apoptosis in breast cancer cells (**Figure 2—figure supplement 1F–H**). Similarly, the staining of α-tubulin in mitotic cells demonstrated more misorientation of the mitotic spindle in CRB3 knockdown groups (**Figure 2E and F**).

In addition to verifying the effect of *Crb3* deletion on promoting proliferation and inhibiting apoptosis in vivo, we detected these markers in primary tumor tissue from PyMT-WT and PyMT-cKO-Crb3 mice. *Crb3* deletion increased the numbers of proliferative and mitotic cells and decreased the numbers of apoptotic cells in primary tumors (**Figure 2G and H**). These results demonstrate that CRB3 could promote the proliferation of mammary epithelial cells, disrupt the mitotic spindle orientation,

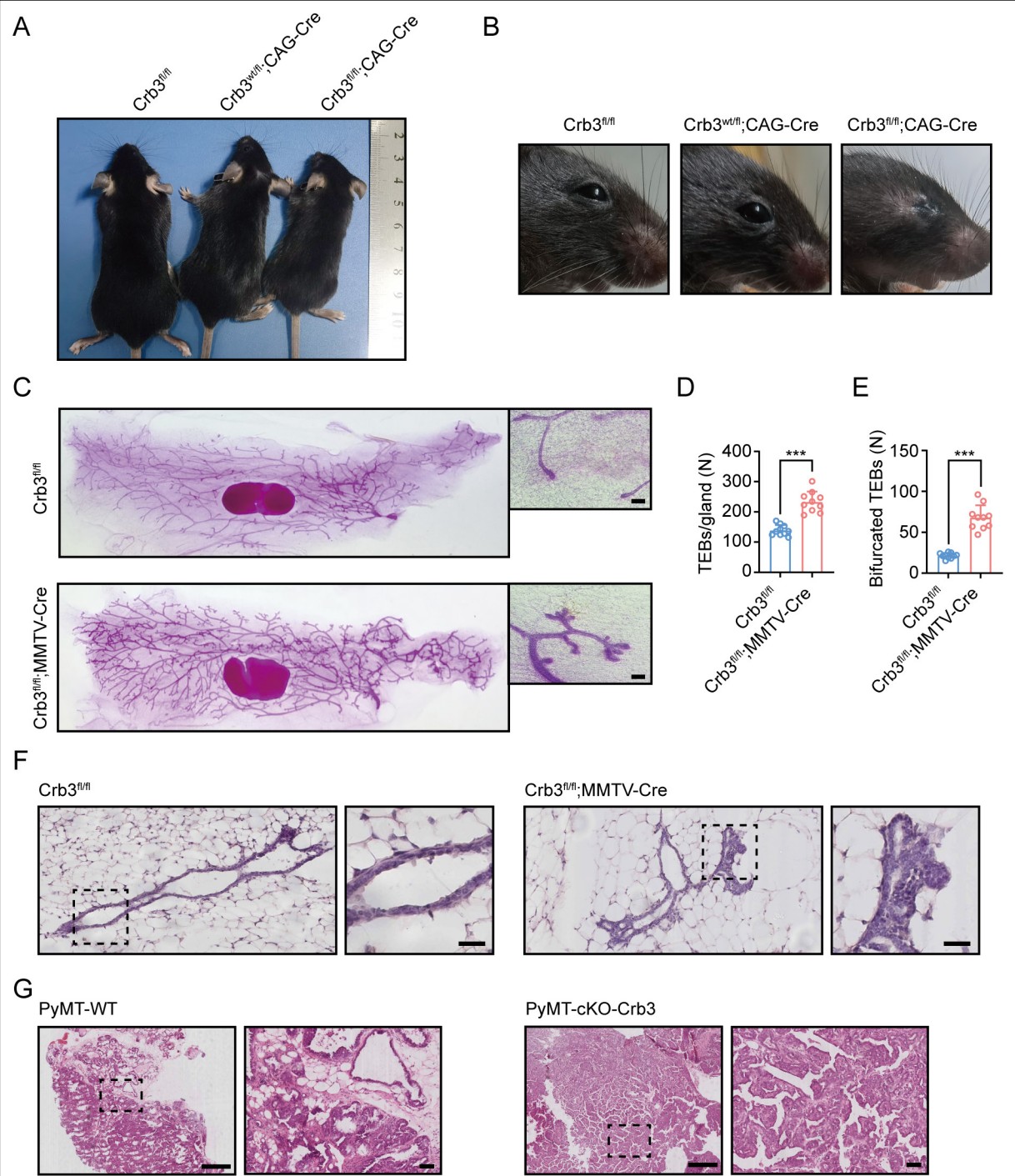

**Figure 1.** *Crb3* knockout mice exhibit smaller sizes and ocular abnormalities, and mammary epithelial cell-specific *Crb3* knockout leads to ductal epithelial hyperplasia and promotes tumorigenesis. (**A**, **B**) Representative whole bodies (**A**) and eyes (**B**) from littermate *Crb3*<sup>fl/fl</sup>, *Crb3*<sup>wt/fl</sup>;CAG-Cre and *Crb3*<sup>fl/fl</sup>;CAG-Cre mice at 4 weeks old. (**C**) Representative mammary whole mounts from littermate *Crb3*<sup>fl/fl</sup> and *Crb3*<sup>fl/fl</sup>;MMTV-Cre mice at 8 weeks old with Carmine-alum staining. (scale bars, 200 μm) (**D**, **E**) Quantification of the average number of terminal end buds (TEBs) and bifurcated TEBs in littermate *Crb3*<sup>fl/fl</sup> (n = 10) and *Crb3*<sup>fl/fl</sup>;MMTV-Cre (n = 10) mice at 8 weeks old. (**F**) Representative images of mammary glands in littermate *Crb3*<sup>fl/fl</sup> and *Crb3*<sup>fl/fl</sup>;MMTV-Cre mice stained with H&E (scale bars, 50 μm). (**G**) Representative images of primary tumors stained with H&E in PyMT-WT and PyMT-cKO-*Crb3* mice at 9 weeks old (scale bars, left 500 μm, right 50 μm). The magnified areas of boxed sections are shown in the right panels. Bars represent means ± SD; unpaired Student's *t*-test, ***p<0.001.

The online version of this article includes the following source data and figure supplement(s) for figure 1:

*Figure 1 continued on next page*

*Figure 1 continued*

**Figure supplement 1.** Generation of *Crb3* knockout mice using the Cre-loxP system.

**Figure supplement 1—source data 1.** Original gel or blot images of *Figure 1—figure supplement 1*.

and protect the internal cells from apoptosis during acini formation, ultimately leading to irregular lumen formation and tumor progression.

## CRB3 regulates primary cilium formation

The primary cilium is an apical antenna-like extensions in epithelial cells that is known to play an important role in controlling lumen formation (*Jonassen et al., 2008*). Previous studies have reported that CRB3 knockdown leads to primary cilium loss in MDCK cells (*Fan et al., 2007*; *Fan et al., 2004*). However, the molecular mechanism that regulates ciliogenesis on apical surfaces is still unclear. To investigate the altered effect of CRB3 on ciliogenesis both in vivo and in vitro, we examined the formation of primary cilia in control MCF10A cells and CRB3 knockdown cells. We found that control MCF10A cells formed primary cilia after becoming confluent, while CRB3 knockdown led to significant ciliogenesis defects (*Figure 3A*). The number of cells with primary cilium was significantly reduced compared to the control group, but the length of the primary cilium that did exist was not different (*Figure 3B*). The overexpression of CRB3b in CRB3-depleted MCF10A cells was able to fully restore the formation of primary cilium to levels comparable to the control group (*Figure 3—figure supplement 1A–C*). Conversely, CRB3b conditional overexpression by adding doxycycline (+dox) resulted in restoring ciliary assembly in MCF7 cells (*Figure 3C* and *Figure 3—figure supplement 1D*). CRB3b overexpression increased the proportion of primary cilium formation in breast cancer cells, and the length of the restored primary cilium was increased (*Figure 3D*). Importantly, the primary cilium could be directly visualized using scanning electron microscopy (SEM) (*Figure 3E*). After CRB3 knockdown, the number of MCF10A cells with primary cilia was significantly decreased, while CRB3b conditional overexpression did not increase these populations (*Figure 3F*).

To further support these findings in vivo, we used breast and kidney tissue to assess ciliogenesis in the *Crb3* knockout mouse model. Renal epithelial cells can form prominent primary cilia, and the absence of primary cilia leads to ciliopathies. We used immunofluorescence to detect ciliary formation in the mammary ductal lumen and renal tubule. Compared to tissues of $Crb3^{fl/fl}$ mice, *Crb3* knockout led to the absence of the primary cilium in the mammary ductal lumen and renal tubule from $Crb3^{fl/fl}$; CAG-Cre mice. Specifically, the proportion of cells forming the primary cilium plummeted (*Figure 3G and H*). This same phenomenon was also observed in MEFs from $Crb3^{fl/fl}$; CAG-Cre mice (*Figure 3—figure supplement 1E and F*). We conclude that CRB3 plays a central role in ciliogenesis in various tissues and mammary cells.

## CRB3 localizes to the basal body of the primary cilium

CRB3, a polarity protein, is typically localized on the tight junctions of the apical epithelium membrane (*Martin-Belmonte and Perez-Moreno, 2011*). Additionally, we noted that CRB3b is found to be closely co-localized with centrosomes at prophase, metaphase, and anaphase (*Fan et al., 2007*). To investigate the relationship between CRB3 and ciliary formation, we first examined CRB3 localization in MCF10A cells expressing CRB3-GFP. In MCF10A cells, exogenous CRB3b was mainly localized at cell tight junctions and could co-localize with pericentrin (a centrosome marker) (*Figure 4A*). Approximately 40% of cells showed this co-localization of exogenous CRB3b and pericentrin (*Figure 4B*). In order to prevent the potential accumulation of exogenously overexpressed CRB3 at the ER and Golgi, which could impact its co-localization with the centrosome, we further verified this phenomenon by observing the localization of endogenous CRB3 (both isoforms CRB3a and CRB3b) with another centrosome marker. Similarly, we detected that CRB3 accumulated on one side of the cytoplasm and had a CRB3 foci located at the basal body represented by γ-tubulin in MCF10A cells (*Figure 4C and D*). Importantly, CRB3 knockdown disturbed the accumulation of γ-tubulin at the basal body in confluent quiescence cells (*Figure 4C and E*). To demonstrate the relationship of this co-localization to the primary cilium, we employed acetylated tubulin to mark the primary cilium. Double immunostaining showed that this CRB3 foci was the basal body of the primary cilium, and CRB3 knockdown inhibited ciliary assembly in MCF10A cells (*Figure 4F and G*). These results suggest that CRB3 is

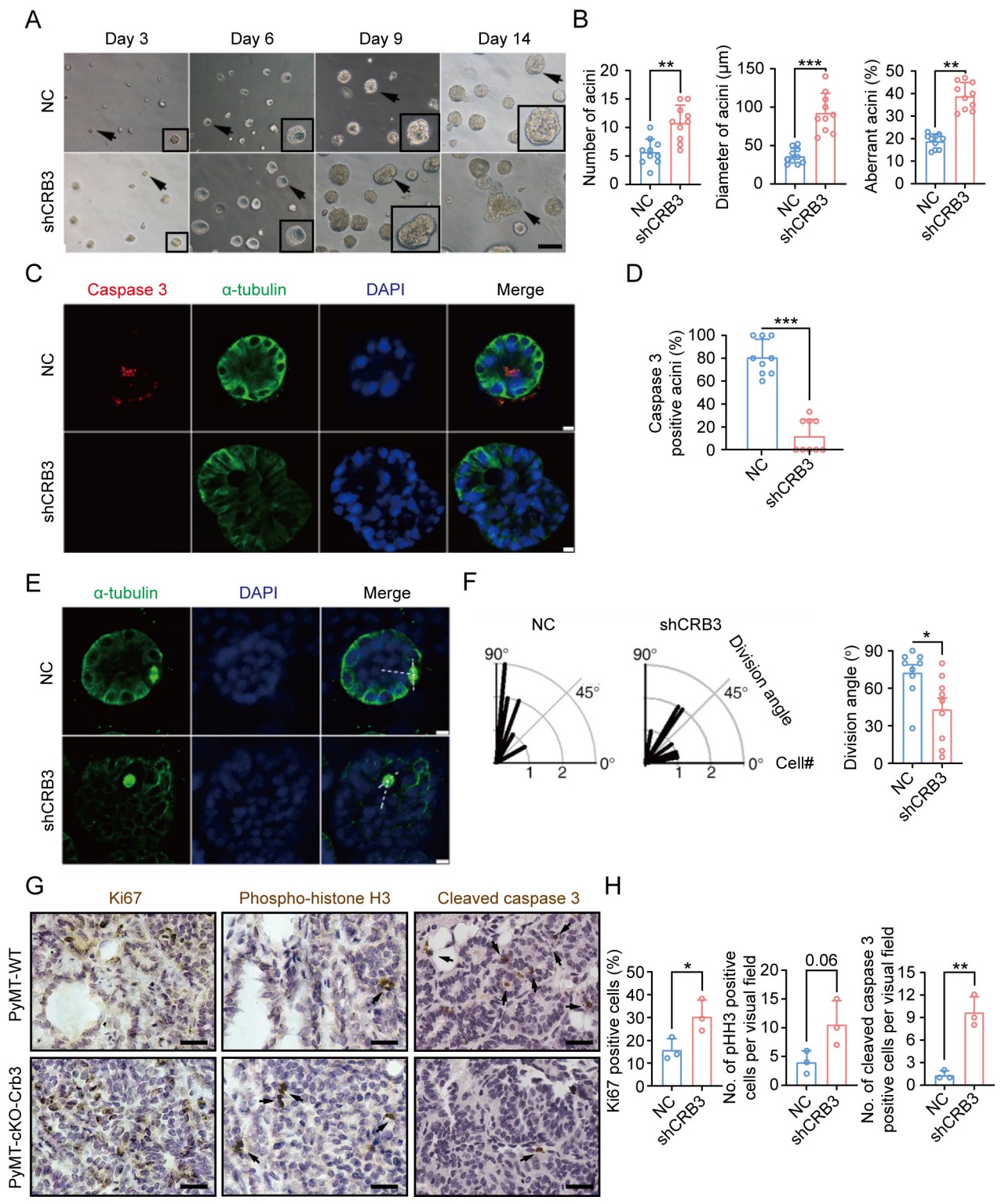

**Figure 2.** CRB3 knockdown inhibits acinar formation of mammary epithelial cells in a 3D culture system. (**A**) Representative effect of CRB3 on acinar formation in the 3D culture system at days 3, 6, 9, and 14. (The magnified areas of the marked arrows are shown in the lower-right corner.) (**B**) Quantification of the average number, diameter, and aberration of acini (at least 3–4 random micrographs were analyzed for each condition in each of three independent experiments, n = 10). (**C**) Immunofluorescence showing apoptosis during lumen formation. Caspase 3 (red), α-tubulin (green), and DNA (blue). (**D**) Quantification of the proportion of caspase 3-positive acini (at least 3–4 random micrographs were analyzed for each condition in each of three independent experiments). (**E**) Immunofluorescence showing the mitotic spindle orientation during lumen formation. α-tubulin (green),

*Figure 2 continued on next page*

*Figure 2 continued*

and DNA (blue). (**F**) Quantification of division angle (at least 3–4 acini were randomly examined for each condition in each of three independent experiments). (**G**) Immunohistochemical analyses of Ki67, phospho-histone H3, and cleaved caspase 3 in primary tumors from PyMT-WT (n = 3) and PyMT-cKO-*Crb3* (n = 3) mice at 9 weeks old. (Positive cells are marked by arrows; scale bars, 25 μm.) (**H**) Quantification of the number of Ki67, phospho-histone H3, and cleaved caspase 3-positive cells from IHC staining images. Bars represent means ± SD; unpaired Student's *t*-test, *p<0.5, **p<0.01, ***p<0.001.

The online version of this article includes the following source data and figure supplement(s) for figure 2:

**Figure supplement 1.** CRB3 knockdown promotes proliferation of mammary epithelial cells, while overexpression promotes apoptosis of breast cancer.

**Figure supplement 1—source data 1.** Original blot images of *Figure 2—figure supplement 1A*.

located at the basal body of the primary cilium and that CRB3 knockdown disturbs the accumulation of γ-tubulin in quiescent cells.

## CRB3 trafficking is mediated by Rab11-positive endosomes

To better investigate the molecular mechanisms of CRB3 in regulating primary cilium formation, we examined some important genes related to ciliogenesis. However, CRB3 knockdown did not affect the mRNA expression of these genes in MCF10A cells (*Figure 5—figure supplement 1A*). Then, we performed coimmunoprecipitation (co-IP) tandem mass spectrometry using tagged exogenous CRB3b as bait to identify its interacting proteins (*Figure 5—figure supplement 1B*). Pathway aggregation analysis of these proteins revealed that these interacting proteins of CRB3b were significantly involved in Golgi vesicle transport and vesicle organization (*Figure 5A*). We focused on these pathways because smaller distal appendage vesicle (DAV) formation is critical for ciliogenesis initiation, and it requires the Rab GTPase Rab11-Rab8 cascade to function (*Lu et al., 2015*). Interestingly, some Rab small GTPase family members identified as CRB3b-binding proteins were aggregated in these pathways, such as Rab10, Rab11A, Rab11B, Rab14, Rab1A, Rab1B, Rab21, Rab2A, Rab32, Rab35, Rab38, Rab5B, Rab5C, and Rab6A. Several centriolar proteins were also identified, including tubulin gamma-1, tubulin gamma-2, CENP-E, CEP290, CEP192, CEP295, and CEP162 (*Figure 5B*).

Rab11-positive vesicles bind to centrosomal Rabin8, leading to ciliary membrane formation (*Westlake et al., 2011*). This finding suggested that the trafficking of Rab11-positive vesicles and polarized vesicles is essential for early ciliary assembly. Next, we examined the polarized vesicle traffic of CRB3 in MCF10A cells. Our studies found that CRB3 could co-localize with EEA1-positive early endosomes, CD63-positive late endosomes, and Rab11-positive recycling endosomes (*Figure 5C*). In addition, after treatment with dynasore, an endocytosis inhibitor, CRB3 significantly accumulated at the cell membrane after 2 hr, and the co-localization of CRB3 with EEA1-, CD63-, and Rab11-positive endosomes was significantly decreased (*Figure 5C and D*). We have previously reported that CRB3 regulates cell contact inhibition, which is significantly downregulated in confluent MCF10A cells (*Mao et al., 2017*). Moreover, dynasore could rescue the CRB3 downregulation in confluent MCF10A cells (*Figure 5E*). The knockdown of CRB3 did not affect Rab11 expression (*Figure 5—figure supplement 1C*). Rab11 knockdown resulted in a predominant localization of CRB3 to the cell membrane and a reduction in intracellular foci (*Figure 5—figure supplement 2A and B*). These results suggest that CRB3 co-localizes with EEA1, CD63, and Rab11 on endosomes, and in particular, Rab11-positive endosomes are involved in the intracellular trafficking of CRB3.

## CRB3 knockdown destabilizes γTuRC assembly during ciliogenesis

According to our findings, we assumed that CRB3 might be involved in polarized vesicle trafficking. The γTuRC is a central regulator of microtubule nucleation and a nucleation site of α-tubulin and β-tubulin at the microtubule-organizing center in mitotic cells (*Kollman et al., 2011*). Additionally, the primary cilium, a microtubule-based structure attached to the basal body, is transformed from the mother centriole. As we found that CRB3 knockdown disturbed the accumulation of γ-tubulin, it prompted us to analyze γTuRC assembly in ciliogenesis. The γTuRC is composed of multiple copies of the γ-tubulin small complex (γTuSC), GCP2 and GCP3, plus GCP4, GCP5, and GCP6 (*Kollman et al., 2011*; *Liu et al., 2020*; *Figure 5F*). Our findings indicate that CRB3 downregulation did not affect the expression of γTuRC subunits (*Figure 5G*). However, we performed co-IPs directed against GCP6 in CRB3-depleted cells and assessed the relative levels of both γTuSC-specific proteins compared

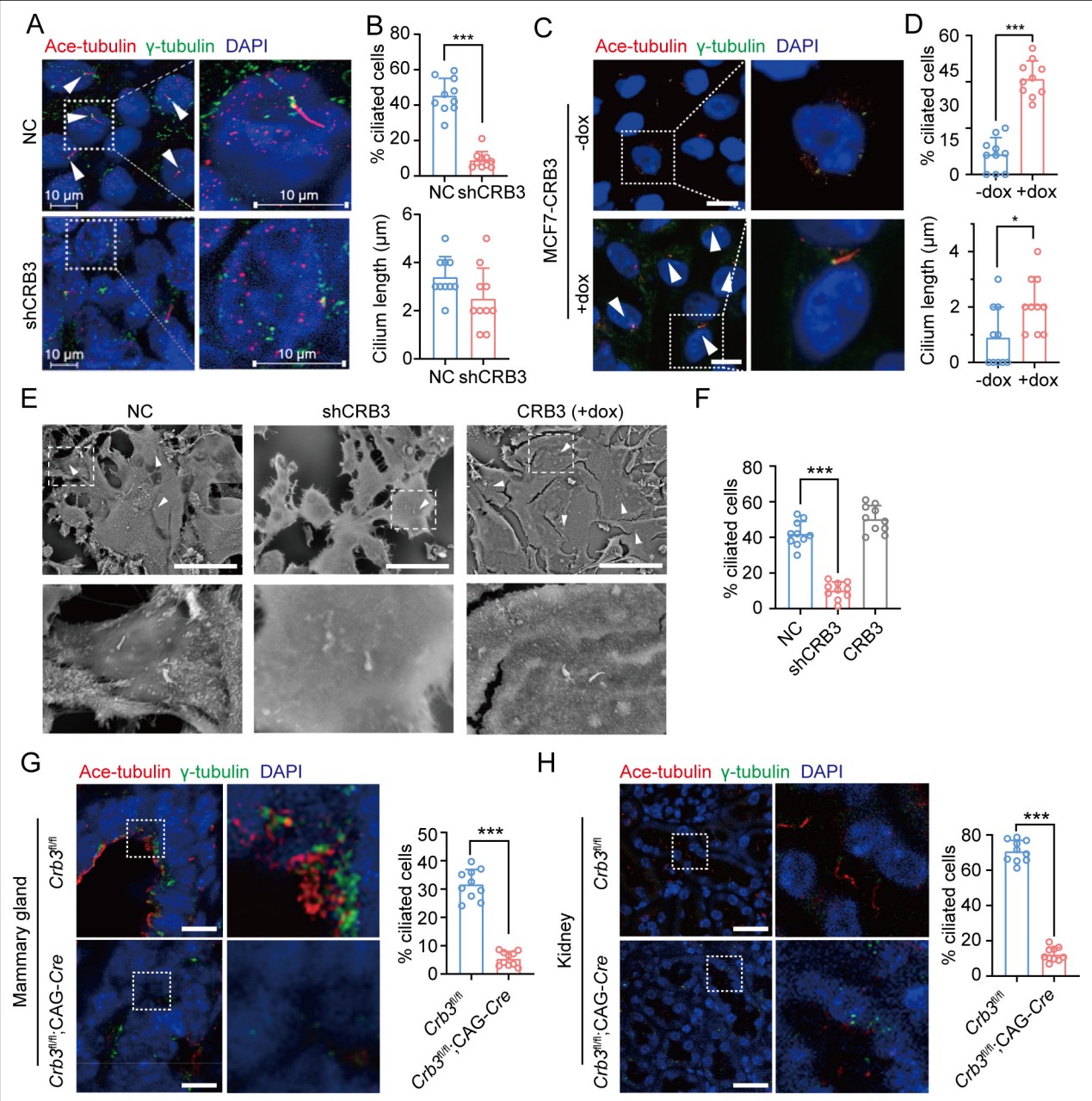

**Figure 3.** CRB3 alters primary cilium formation in mammary cells, mammary ductal lumen, and renal tubule from *Crb3*fl/fl;CAG-Cre mice. (**A**, **C**) Representative images of immunofluorescent staining of primary cilium formation with CRB3 knockdown in MCF10A cells and CRB3b conditional overexpression upon dox induction in MCF7 cells. Acetylated tubulin (red), γ-tubulin (green), and DNA (blue). (The primary cilium is marked by arrows; scale bars, 10 μm.) (**B**, **D**) Quantification of the proportion of cells with primary cilium formation and the length of the primary cilium (at least 3–4 random micrographs were analyzed for each condition in each of three independent experiments, n = 10). (**E**) Representative scanning electron microscope images of primary cilium formation with CRB3 knockdown and CRB3b conditional overexpression in MCF10A cells. (The primary cilium is marked by arrows; scale bars, 50 μm.) (**F**) Quantification of the proportion of cells with primary cilium formation (at least 3–4 random micrographs were analyzed for each condition in each of three independent experiments, n = 10). (**G**, **H**) Representative immunofluorescent staining of primary cilium formation in the mammary ductal lumen and renal tubule from *Crb3*fl/fl (n = 10) and *Crb3*fl/fl;CAG-Cre (n = 10) mice, respectively. Acetylated tubulin (red), γ-tubulin (green), and DNA (blue) (scale bars, 25 μm). Bars represent means ± SD; unpaired Student's *t*-test, *p<0.05, ***p<0.001.

The online version of this article includes the following source data and figure supplement(s) for figure 3:

**Figure supplement 1.** CRB3 alters primary cilium formation in MCF10A cells and mouse embryonic fibroblasts (MEFs) from *Crb3*fl/fl;CAG-Cre mice.

**Figure supplement 1—source data 1.** Original blot images of *Figure 3—figure supplement 1*.

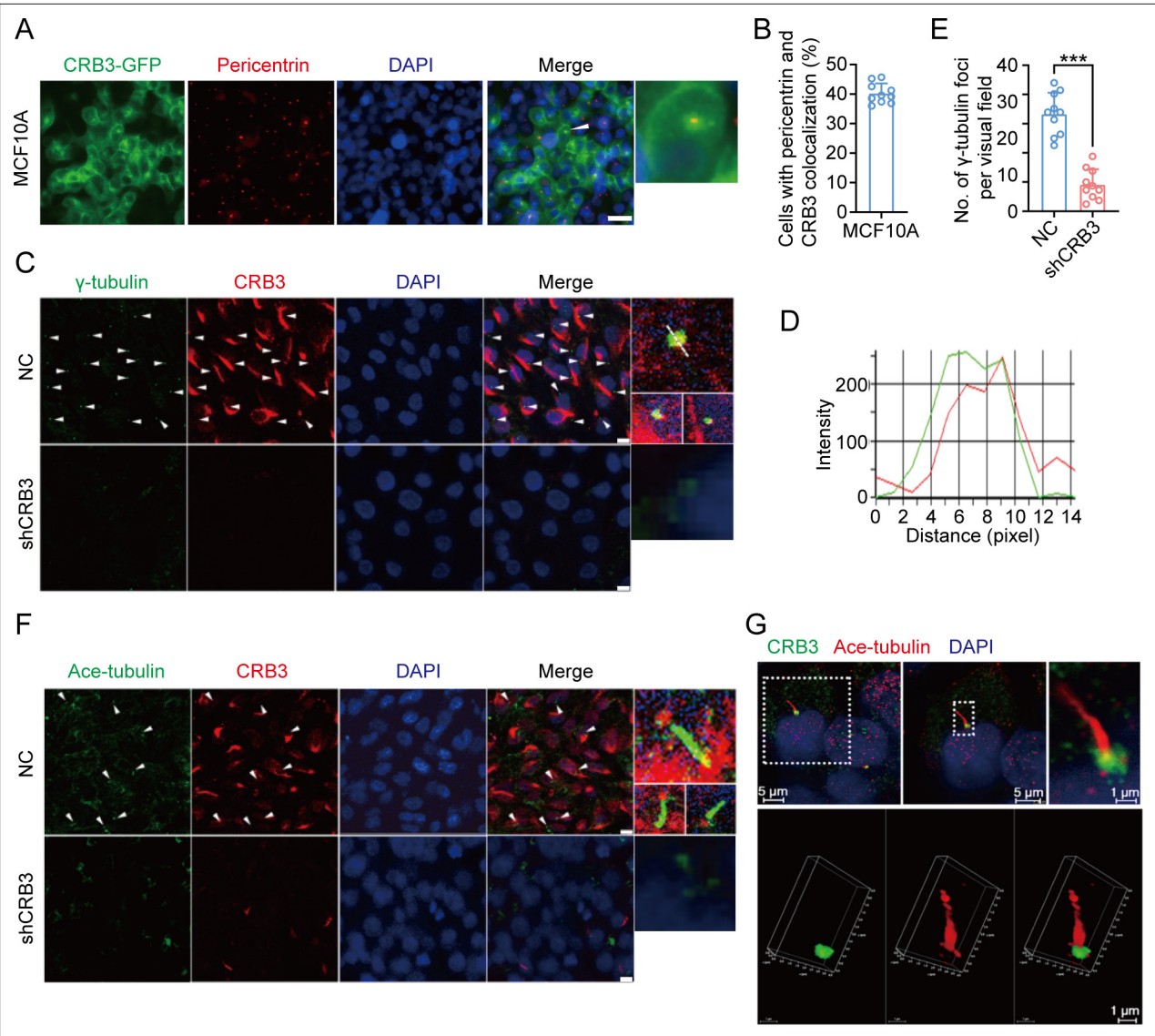

**Figure 4.** CRB3 localizes to the basal body of the primary cilium. (**A**) Immunofluorescence showing the co-localization of exogenous CRB3b with centrosomes in MCF10A cells. Pericentrin, a marker of centrosome (red), CRB3-GFP (green), and DNA (blue) (co-localization is marked by arrows; scale bars, 10 μm). (**B**) Quantification of the proportion of cells with pericentrin and exogenous CRB3b co-localization (at least 3–4 random micrographs were analyzed for each condition in each of three independent experiments, n = 10). (**C**) Another co-localization of endogenous CRB3 with the basal body in MCF10A cells. γ-Tubulin is a marker of the centrosome and basal body of the primary cilium (green), CRB3 (red), and DNA (blue) (co-localization is marked by arrows; scale bars, 10 μm). (**D**) Corresponding fluorescence intensity profile across a section of the array, as indicated by the dashed white line in (**C**). (**E**) Quantification of the number of γ-tubulin foci in MCF10A cells from IF staining images (at least 3–4 random micrographs were analyzed for each condition in each of three independent experiments, n = 10). (**F**) Double immunostaining displaying the co-localization of CRB3 with the primary cilium in MCF10A cells. Acetylated tubulin (green), CRB3 (red), and DNA (blue) (co-localization is marked by arrows; scale bars, 10 μm). (**G**) Fluorescence 3D reconstruction of CRB3 and primary cilium co-localization. Acetylated tubulin (red), CRB3 (green), and DNA (blue). Bars represent means ± SD; unpaired Student's *t*-test, ***p<0.001.

with the control shRNA-treated cells. Interestingly, CRB3 knockdown significantly reduced the interaction of GCP6 with GCP3 (*Figure 5H*). To verify this phenomenon, we observed the localization of GCP3 with GCP6 foci in MCF10A cells. Correspondingly, CRB3 downregulation significantly disturbed the co-localization of GCP3 with GCP6 foci in quiescence cells (*Figure 5I*). To investigate whether CRB3 facilitates γTuRC assembly, we fractionated MCF10A cytoplasmic proteins on sucrose gradients according to published methods (*Farache et al., 2016*). We noticed that γTuSC were sedimented mainly in fractions 3, while γTuRC sedimented in fractions 6. However, CRB3 downregulation caused most of GCP6 and GCP5 to be precipitated independently of γTuRC in the low-density fractions

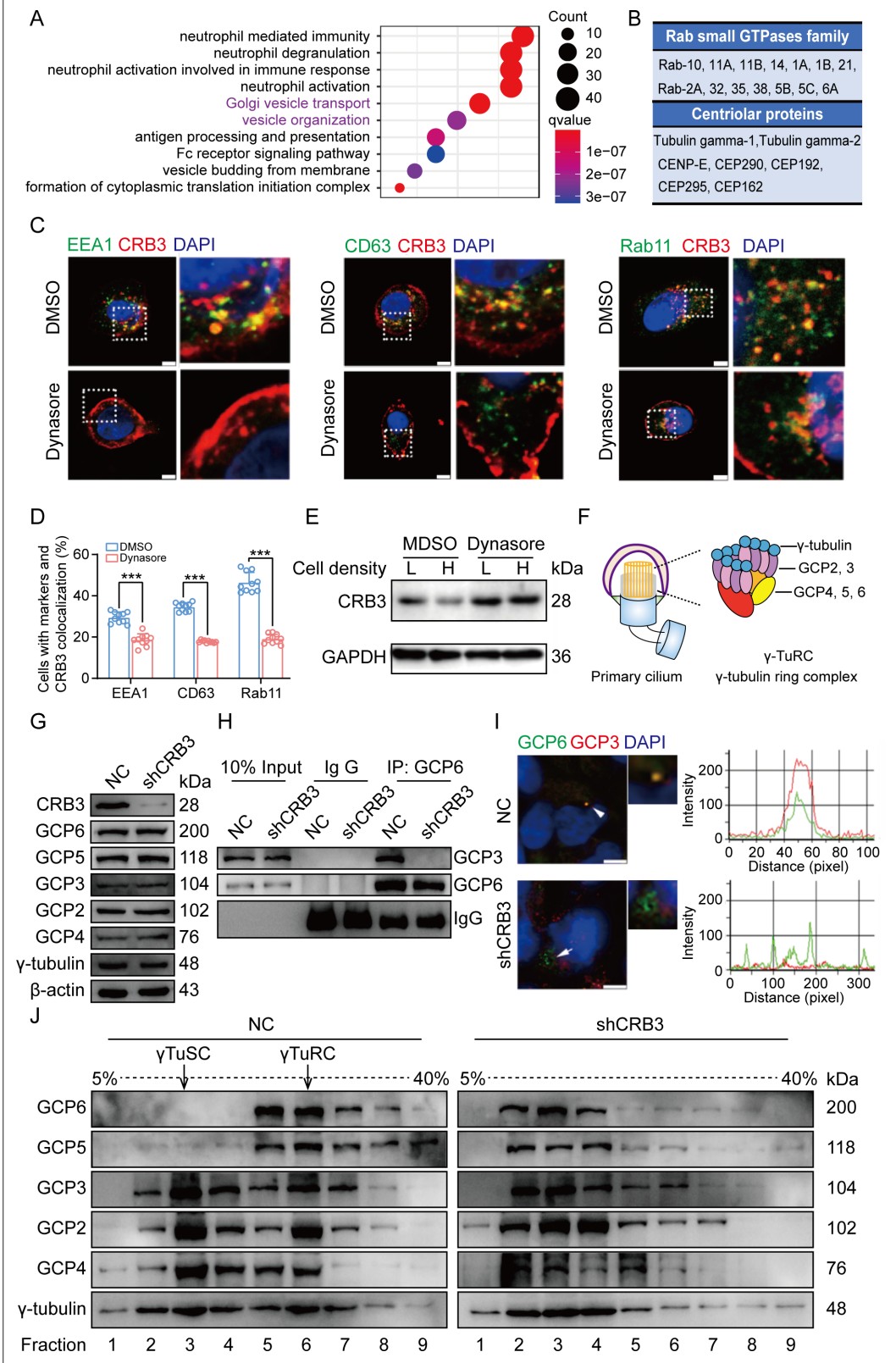

**Figure 5.** CRB3 trafficking is mediated by Rab11-positive endosomes, and CRB3 knockdown destabilizes γ-tubulin ring complex (γTuRC) assembly during ciliogenesis. (**A**) Pathway aggregation analysis of CRB3b-binding proteins identified by mass spectrometry in MCF10A cells. (**B**) Table of some Rab small GTPase family members and centriolar proteins identified as CRB3b-binding proteins. (**C**) Immunofluorescence showing the co-localization

*Figure 5 continued on next page*

*Figure 5 continued*

of CRB3 with EEA1-, CD63-, and Rab11-positive endosomes in MCF10A cells. EEA1, CD63, Rab11 (green), CRB3 (red), and DNA (blue) (scale bars, 10 µm). (**D**) Quantification of the proportion of cells with these markers and CRB3 co-localization (n = 10). (**E**) Western blotting showing the levels of CRB3 in MCF10A cells treated with dynasore at different cell densities. (**F**) The structure diagram of γTuRC. (**G**) Immunoblot analysis of the effect of CRB3 on γTuRC molecules in MCF10A cells. (**H**) Coimmunoprecipitation showing the interacting proteins with GCP6 in MCF10A cells with CRB3 knockdown. (**I**) Representative images of immunofluorescent staining of GCP3 and GCP6 co-localization in MCF10A cells with the corresponding fluorescence intensity profile. GCP3 (red), GCP6 (green), and DNA (blue) (at least 3–4 random micrographs were analyzed for each condition in each of three independent experiments, scale bars, 10 µm). (**J**) The comparison of cytoplasmic extracts from MCF10A cells and cells with CRB3 knockdown after fractionation in sucrose gradients. The γ-tubulin small complex (γTuSC) sedimentation was mainly in fractions 3, and γTuRC sedimentation was mainly in fractions 6. Bars represent means ± SD; unpaired Student's *t*-test, ***p<0.001.

The online version of this article includes the following source data and figure supplement(s) for figure 5:

**Source data 1.** Raw data for the identification of CRB3b-binding proteins by tandem mass spectrometry related to *Figure 5A and B*.

**Source data 2.** Original blot images of *Figure 5*.

**Figure supplement 1.** The effect of CRB3 on the expression of some ciliogenesis-related genes and Rab11.

**Figure supplement 1—source data 1.** Original gel or blot images of *Figure 5—figure supplement 1*.

**Figure supplement 2.** Rab11 knockdown caused alterations in CRB3 trafficking and defects in primary cilium.

**Figure supplement 2—source data 1.** Original blot images of *Figure 5—figure supplement 2A*.

(*Figure 5J*). Together, these results demonstrate that the depletion of CRB3 disrupts molecular interactions between the γTuRC proteins, resulting in a failure of microtubule formation in the primary cilium. However, this depletion may not affect the expression of these molecules during γTuRC assembly. Based on these findings, we hypothesize that CRB3 could facilitate the polarized vesicle trafficking of some γTuRC-specific proteins through Rab11-positive endosomes.

## CRB3 interacts with Rab11

To fully prove this hypothesis, we first examined the interaction of CRB3 with Rab11-positive endosomes and γTuRC-specific proteins in MCF10A cells. Using CRB3 as bait, we identified that CRB3 could interact with Rab11, but not GCP6 and GCP3 (*Figure 6A*). In contrast, Rab11 could interact with CRB3 and GCP6 instead of GCP3 by tagging Rab11 as bait (*Figure 6B*). The co-IP results showed that CRB3 knockdown did not affect Rab11 interacting with GCP6, but the level of Rab11 binding to GCP6 tended to decrease in MCF10A cells with CRB3 knockdown (*Figure 6C*). Consistent with other reports (*Knödler et al., 2010*), Rab11 knockdown significantly inhibited primary cilium formation in MCF10A cells (*Figure 5—figure supplement 2C and D*).

Based on these results, we further detected the region of CRB3b that interacts with Rab11 by using coexpression and co-IP in HEK293 cells. CRB3b is composed of a signal peptide (SP), extracellular domain, transmembrane (TM), FERM-binding domain (FBD), and carboxy-terminal PDZ-binding domain (PBD). We constructed CRB3-GFP fusion proteins by fusing GFP tags to the extracellular domain, as previously reported (*Djuric et al., 2016*; *Figure 6D*). According to these domains, we also constructed the truncations of CRB3-GFP fusion proteins with serial C-terminal deletions (*Figure 6E*) and another Flag-Rab11a fusion protein. The coexpression and co-IP results showed that Flag-Rab11a interacts with full-length CRB3 (1–120), CRB3 (1–116), CRB3 (1–83), and CRB3 (1–58) but not CRB3 (1–26) (*Figure 6F*). Therefore, the region of CRB3b that interacts with Rab11 is amino acids 27–58.

Rab11 is a small GTP-binding protein that plays an essential role in regulating the dynamics of recycling endosomes, which must undergo GDP/GTP cycles. In particular, Rab11a[S25N] is a GDP-locked form, while Rab11a[S20V] and Rab11a[Q70L] are GTP-locked forms (*Knödler et al., 2010*). To further investigate the interaction between the GTP-bound form of Rab11a and CRB3b, we constructed these mutant variants of Rab11a. Next, we found that CRB3b interacts more strongly with Rab11a[Q70L] and Rab11a[S20V], while weakly interacting with Rab11a[S25N], compared to Rab11aWT (*Figure 6G*). Together, these results indicate that the interaction of CRB3b and Rab11a is dependent on the GTPase activity of Rab11a.

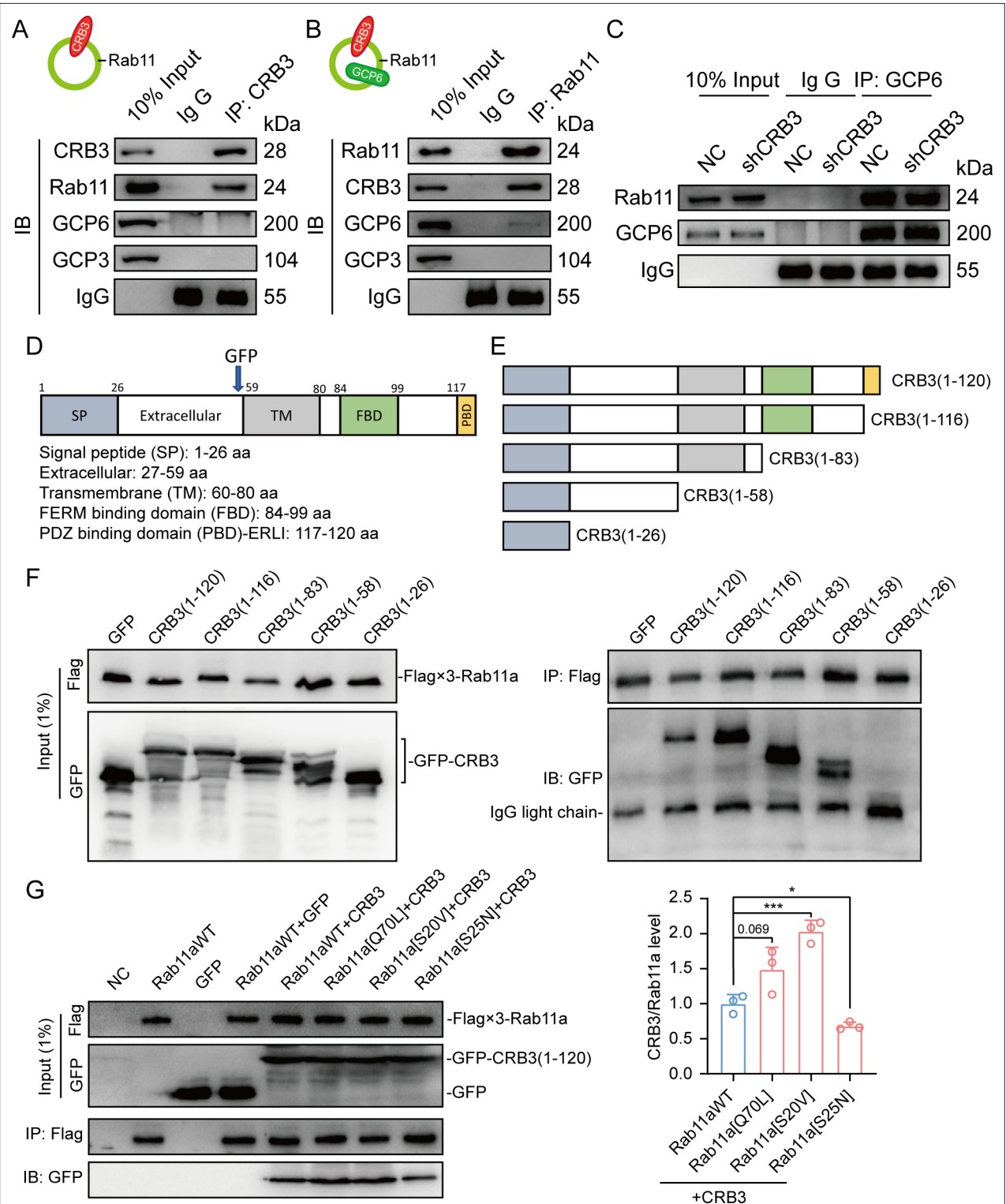

**Figure 6.** CRB3 interacts with Rab11. (**A**) Coimmunoprecipitation of CRB3 with Rab11, GCP6, and GCP3 in MCF10A cells. (**B**) Coimmunoprecipitation of Rab11 with CRB3, GCP6, and GCP3 in MCF10A cells. (**C**) Coimmunoprecipitation of Rab11 with GCP6 in control and CRB3 knockdown MCF10A cells. (**D**) Schematic diagram of CRB3b domains. (**E**) Diagram truncations of CRB3b-GFP fusion proteins with serial C-terminal deletions. (**F**) Domain mapping of CRB3b-GFP for Flag-Rab11a binding. Flag antibody co-IP of the full-length CRB3b-GFP and truncations of CRB3b-GFP with Flag-Rab11a were cotransfected into HEK293 cells for 48 hr. Immunoblot analysis was performed using GFP and Flag antibodies. (**G**) Coimmunoprecipitation of Rab11 mutant variants with full-length CRB3b-GFP. Flag antibody co-IP of the full-length CRB3b-GFP with Flag-Rab11aWT, Flag-Rab11a[Q70L],

*Figure 6 continued on next page*

*Figure 6 continued*

Flag-Rab11a[S20V], and Flag-Rab11a[S25N] were cotransfected into HEK293 cells for 48 hr. Immunoblot analysis was performed using GFP and Flag antibodies. Bars represent means ± SD, and the experiments were performed in triplicate; unpaired Student's *t*-test, *p<0.05, ***p<0.001.

The online version of this article includes the following source data for figure 6:

**Source data 1.** Original blot images of *Figure 6*.

## CRB3 navigates GCP6/Rab11 trafficking vesicles to CEP290 in the primary cilium

Since CRB3 knockdown does not affect the interaction with Rab11-positive endosomes and γTuRC-specific proteins GCP6, our focus was to determine whether CRB3 affects the localization of GCP6/Rab11 trafficking vesicles to the basal body of the primary cilium. We reviewed the identification of CRB3 interacting proteins and found some centriolar proteins. CEP290, for example, is located in the transition zone of the primary cilium and is required for the formation of microtubule-membrane linkers (*Craige et al., 2010*). Then, we verified that exogenous CRB3b could bind to CEP290, Rab11, GCP6 (*Figure 7A*), and Rab11 also interacted with CEP290, CRB3-GFP, and GCP6 (*Figure 7B*). Similar to *Figure 6C*, the amount of GCP6 bound to Rab11 was reduced within CRB3 knockdown. Importantly, CRB3 knockdown made it difficult for Rab11 to bind CEP290 (*Figure 7C*). We corroborated these findings by checking the co-localization of the basal body foci of GCP6 and γ-tubulin, γ-tubulin and Rab11 in MCF10A and MEF cells. In quiescence control cells, GCP6 foci and γ-tubulin formed a prominent focus at the basal body of the primary cilium, as observed by immunofluorescence. However, CRB3 downregulation eliminated this focus and significantly disturbed the basal body foci of GCP6 and γ-tubulin in MCF10A cells (*Figure 7D*), and γ-tubulin and Rab11 in MEF cells (*Figure 7E*). At the beginning of this experiment, we also attempted to show the interaction of CRB3, Rab11, and GCP6 by expressing fluorescence-tagged GCP6 protein and using FRET assay. However, the centrosomal foci of fluorescent GCP6 was difficult to capture due to weak signal, and this may require re-overexpression of ninein-like protein (Nlp) to promote γTuRC enrichment (*Farache et al., 2016*; *Casenghi et al., 2003*).

Although CRB3 knockdown had little effect on the formation of GCP6/Rab11 trafficking vesicles, it significantly inhibited γTuRC assembly during ciliogenesis. This, in turn, impacted the transport of GCP6/Rab11 trafficking vesicles to the basal body of the primary cilium. Since CRB3 affected the binding of Rab11 and CEP290, we speculate that CRB3 may act as a navigator, navigating GCP6/Rab11 trafficking vesicles to the basal body of the primary cilium.

## CRB3 regulates the Hh and Wnt signaling pathways in tumorigenesis

To investigate the role of CRB3 in regulating ciliary assembly in breast cancer, we examined the relationship between CRB3 and the primary cilium in breast cancer tissues. We found that CRB3 was localized at the subapical surface of the mammary gland lumen in the paracarcinoma tissues, but no obvious expression or localization in breast cancer tissues (*Figure 8A*). Immunofluorescence analysis showed that the intact primary cilium was observed in the adjacent para-carcinoma tissues, whereas the number of cells with primary cilia was much lower in breast cancer tissues (*Figure 8B and C*). Consistent with previous findings, CRB3 was localized at the basal body of the primary cilium in adjacent para-carcinoma tissues (*Figure 8D*). These results again suggest that a defect in CRB3 expression inhibits ciliary assembly, which could be involved in tumorigenesis.

To investigate the effects of primary cilium absence, we examined alterations in the Hh and Wnt signaling pathways in CRB3-depleted cells. Firstly, we detected the translocation of Smoothened (SMO) into the primary cilium and the expression of the target gene GLI1. SAG, a potent SMO receptor agonist, activates the Hh signaling pathway. However, in MCF10A cells treated with SAG, CRB3 knockdown did not promote SMO translocation into the primary cilium (*Figure 8E*). After SAG treatment, the mRNA expression of GLI1 was significantly decreased in CRB3-depleted cells (*Figure 8F*). Crb3 knockout did not alter GLI1 expression and nuclear localization in primary tumors of PyMT-cKO-Crb3 mice (*Figure 8H*). We previously reported that CRB3 expression was high in immortalized mammary epithelial cells, whereas loss of CRB3 expression was observed in breast cancer cells (*Mao et al., 2017*). On the other hand, western blotting showed that CRB3 knockdown resulted in significant downregulation of GSK3-β in MCF10A cells. Additionally, β-catenin, the major effector of

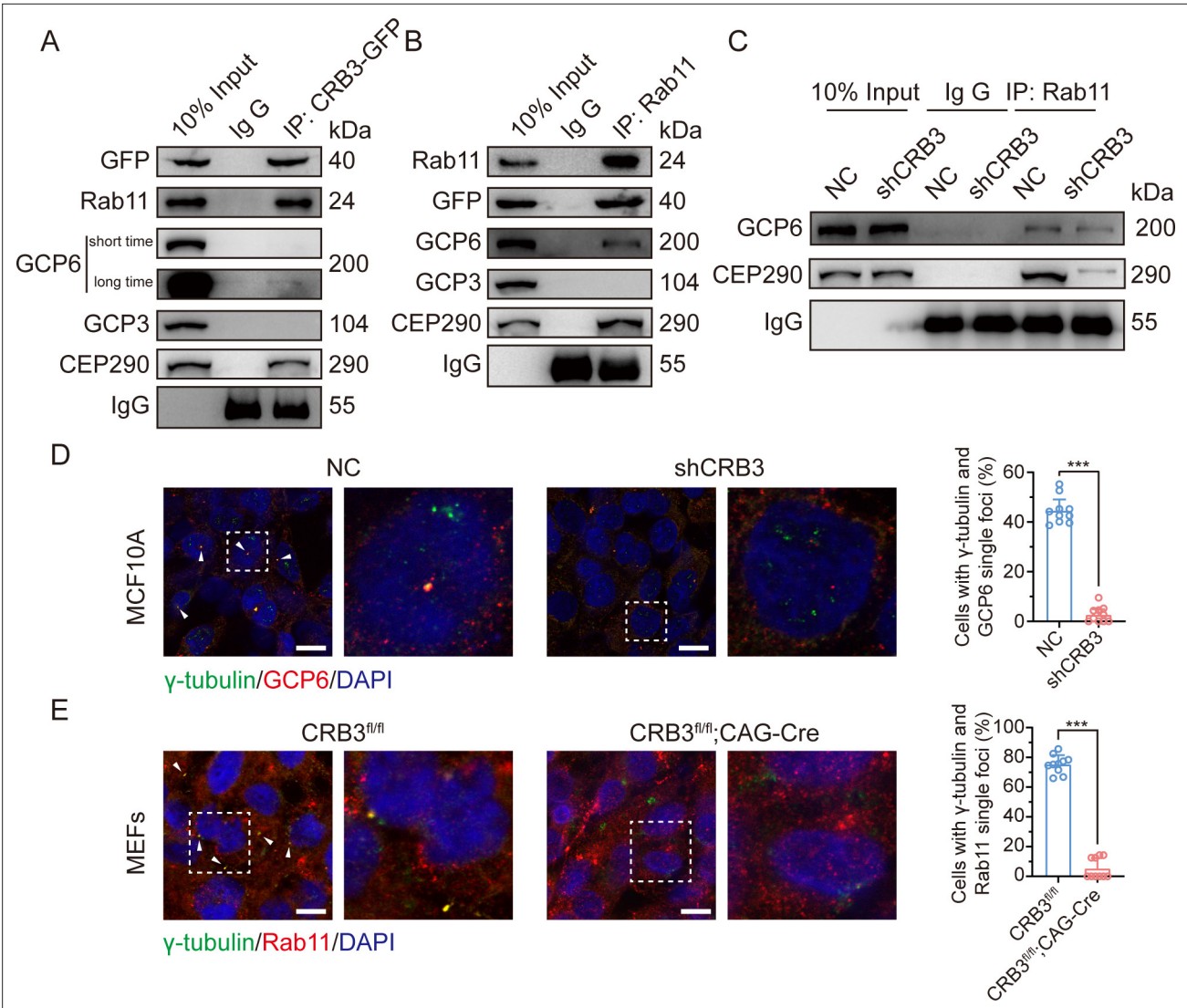

**Figure 7.** CRB3 navigates GCP6/Rab11 trafficking vesicles to the basal body of the primary cilium. (**A**) Coimmunoprecipitation of exogenous CRB3b with Rab11, GCP6, GCP3, and CEP290 in MCF10A cells. (**B**) Coimmunoprecipitation of Rab11 with exogenous CRB3b, GCP6, GCP3, and CEP290 in MCF10A cells. (**C**) Coimmunoprecipitation of Rab11 with GCP6 and CEP290 in control and CRB3 knockdown MCF10A cells. (**D**) Representative immunofluorescent images of GCP6 and γ-tubulin co-localization of the basal body foci in MCF10A cells with CRB3 knockdown. γ-Tubulin (green), GCP6 (red), and DNA (blue) (co-localization is marked by arrows; scale bars, 25 μm). (**E**) Representative immunofluorescent images of Rab11 and γ-tubulin co-localization of the basal body foci in mouse embryonic fibroblast (MEF) cells from *Crb3*^fl/fl and *Crb3*^fl/fl;CAG-Cre mice. γ-Tubulin (green), Rab11 (red), and DNA (blue). Foci are marked by arrows; scale bars, 25 μm. Bars represent means ± SD; unpaired Student's *t*-test, ***p<0.001.

The online version of this article includes the following source data for figure 7:

**Source data 1.** Original blot images of *Figure 7*.

the canonical Wnt signaling pathway, was upregulated. In breast cancer cells, CRB3b overexpression led to upregulation of GSK3-β and downregulation of β-catenin (*Figure 8G* and *Figure 8—figure supplement 1*). Notably, β-catenin was significantly upregulated and markedly localized in the nucleus of PyMT-cKO-Crb3 mice (*Figure 8I*). Overall, these data suggest that CRB3 knockdown cannot activate the Hh signaling pathway but can activate the Wnt signaling pathway, which ultimately contributes to tumorigenesis.

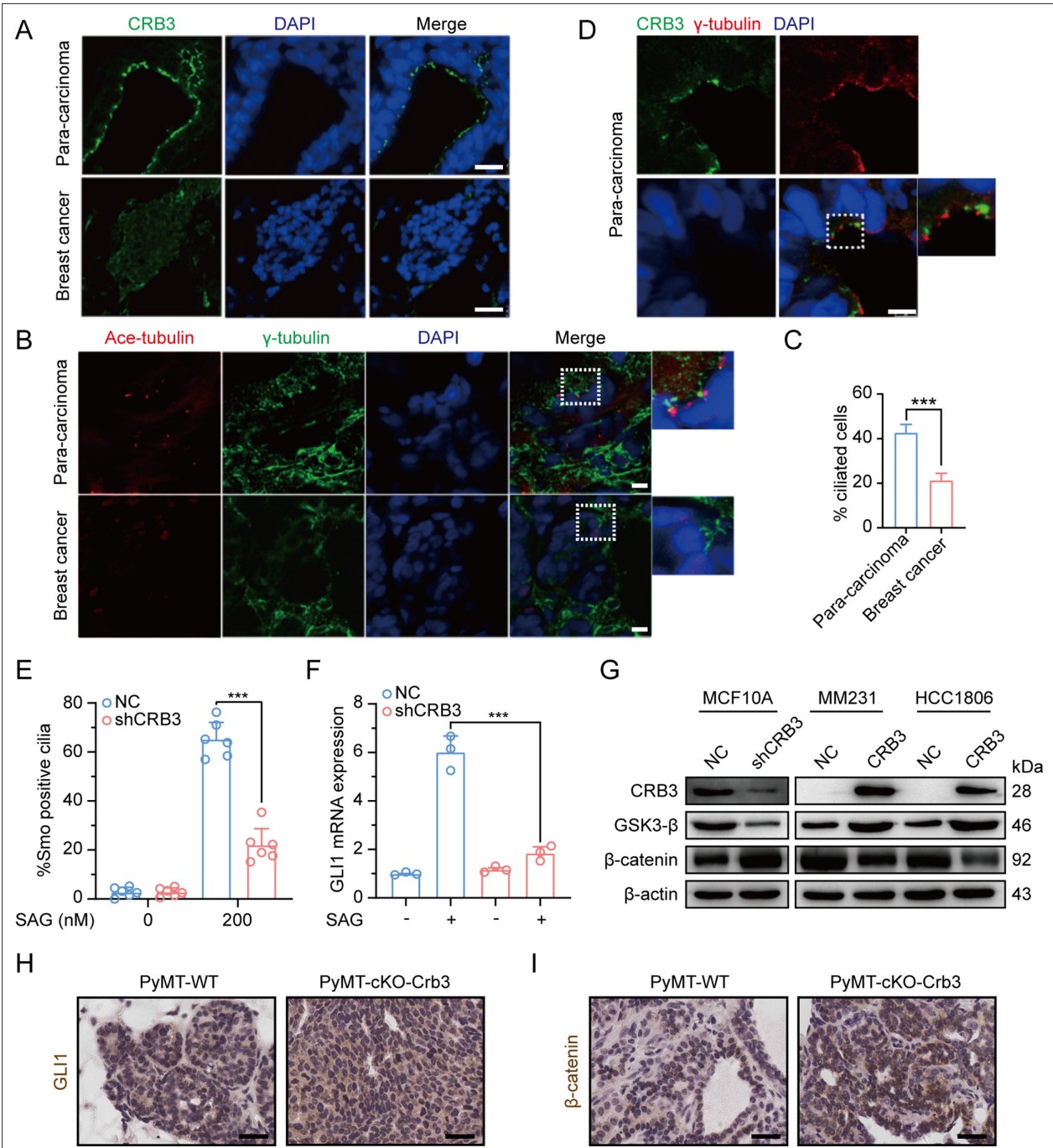

**Figure 8.** Defects in CRB3 expression inhibit ciliary assembly in breast cancer tissues and activate the Wnt signaling pathway in mammary cells and PyMT mouse model. (**A**) Representative immunofluorescent images of CRB3 in breast cancer tissues (n = 50). CRB3 (green) and DNA (blue) (scale bars, 25 μm). (**B**) Representative immunofluorescent images of the primary cilium in breast cancer tissues (n = 50). Acetylated tubulin (red), γ-tubulin (green), and DNA (blue) (scale bars, 10 μm). (**C**) Quantification of the proportion of cells with primary cilium formation in breast cancer tissues (n = 50). (**D**) Representative immunofluorescent images of CRB3 and γ-tubulin co-localization in adjacent paracarcinoma tissues. CRB3 (green), γ-tubulin (red), and DNA (blue) (scale bars, 25 μm). (**E**) Quantification of the proportion of MFC10A cells with SMO translocation after CRB3 knockdown (n = 6). (**F**) Real-time quantitative PCR showing the relative mRNA expression of GLI1 upon SAG treatment in CRB3-depleted MFC10A cells (n = 6). (**G**) Immunoblot analyses of the effect of CRB3 on the molecules of the Wnt signaling pathway in mammary cells, and the experiments were performed in triplicate. (**H. I**) Immunohistochemical analyses of GLI1 and β-catenin in primary tumors from PyMT-WT and PyMT-cKO-*Crb3* mice at 9 weeks old, respectively (scale bars, 25 μm). Bars represent means ± SD; unpaired Student's *t*-test, ***p<0.001.

The online version of this article includes the following source data and figure supplement(s) for figure 8:

*Figure 8 continued on next page*

*Figure 8 continued*

**Source data 1.** Original blot images of *Figure 8*.

**Figure supplement 1.** Western blotting quantification of the molecules in the Wnt signaling pathway.

## Discussion

Apical-basal cell polarity is essential for cellular architecture, development, and homeostasis in epithelial tissues. However, most malignant cancers lack cell polarity due to the unconstrained cell cycle and cell migration with decreased adhesion. CRB3, an important apical protein, is significantly reduced in various tumor cells and tissues, as well as promotes metastasis and tumor formation in nude mice (*Li et al., 2017*; *Mao et al., 2017*; *Karp et al., 2008*; *Li et al., 2018*; *Thiery et al., 2009*; *Varelas et al., 2010*). Here, we generated a novel transgenic mouse model with conditional deletion of Crb3 to investigate the role of polarity proteins in tumorigenesis. *Crb3* knockout mice were reported to have cystic kidneys, improper airway clearance in the lung, villus fusion, apical membrane blebs, and disrupted microvilli in intestine airway clearances, which demonstrated that CRB3 is crucial for the development of the apical membrane and epithelial morphogenesis (*Whiteman et al., 2014*; *Charrier et al., 2015*). Importantly, our studies in the *Crb3* knockout mouse model suggest that in addition to death after birth, it is characterized by smaller size, ocular abnormality, bronchial smooth muscle layer defect, reduced pancreatic islets, and hyperplasia mammary ductal and renal tubular epithelium (some data not shown). These important phenotypic changes are due to the disruption of epithelial polarity homeostasis, which may ultimately lead to tumorigenesis.

In addition, we observed the absence of primary cilia in the mammary ductal lumen and renal tubule in the *Crb3* knockout mouse model. Consistent with other reports, CRB3 knockdown regulated ciliary assembly in MDCK cells (*Fan et al., 2007*; *Fan et al., 2004*). Furthermore, CRB3 was localized in the inner segments of photoreceptor cells and concentrated with the connecting cilium throughout the development of the mouse retina (*Herranz-Martín et al., 2012*). These ocular defect phenotypes closely resemble those of EHD1 knockout mice, which regulate ciliary vesicle formation in primary cilium assembly (*Lu et al., 2015*; *Arya et al., 2015*; *Rainey et al., 2010*). Hence, CRB3 can affect primary cilium formation in various epithelial tissues, such as the breast, kidney, and retina.

Previous literature has reported that CRB3 localizes within the primary cilium in differentiated MDCK cells (*Fan et al., 2007*; *Fan et al., 2004*). However, our findings indicate that it is mainly located on the basal body of the primary cilium in mammary epithelial cells. The immunofluorescence results in these literatures showed that CRB3 was scattered on the primary cilia, with a strong foci at the basal body. Notably, in rat kidney-collecting ducts, the localization of CRB3 on primary cilia was significantly reduced, while the obvious localization was at the basal body (*Fan et al., 2004*). Another study also reported the co-localization of CRB3b and γ-tubulin in MDCK cells (*Fan et al., 2007*). This finding is consistent with our conclusion, and we also verified its co-localization with the centrosome by overexpressing CRB3b in mammary epithelial cells, indicating that CRB3 mainly localizes to the basal body of the primary cilium. Since CRB3b also alters primary cilium formation in the mammary epithelium, it is important to investigate the molecular mechanisms by which CRB3 regulates primary cilium formation. Subsequently, we used mass spectrometry to identify the CRB3b-binding protein. Our findings indicate that CRB3b not only binds to some centrosomal proteins, but also plays a significant role in Golgi vesicle transport (*Figure 5A and B*). These results are consistent with the results of the endogenous CRB3 staining by laser confocal microscopy (*Figure 4C*). Specifically, endogenous CRB3 localizes to the centrosome and strongly accumulates in an undefined intracellular structure near the cell nucleus, likely to be the Golgi. However, when we compared the fluorescence microscopy analysis of exogenous CRB3b staining (*Figure 4A*), we found that the main localization was observed at the cell tight junctions, which is consistent with the reported molecular function. Additionally, there was a strong accumulation of staining around the cell nucleus, with a shape different from the endogenous staining around the cell nucleus. We speculate that this may be due to excessive overexpression of exogenous proteins, which obstruct the protein secretion pathway and result in the accumulation of the exogenous protein at the ER and Golgi. To better study the cell localization of exogenous CRB3b, a conditional induction expression or knock-in tag on the endogenous CRB3 gene could be used.

Polarized vesicular traffic plays a crucial role in extending the primary cilium on apical epithelial surfaces. However, the molecular mechanism and its relationship to tumorigenesis remain unclear.

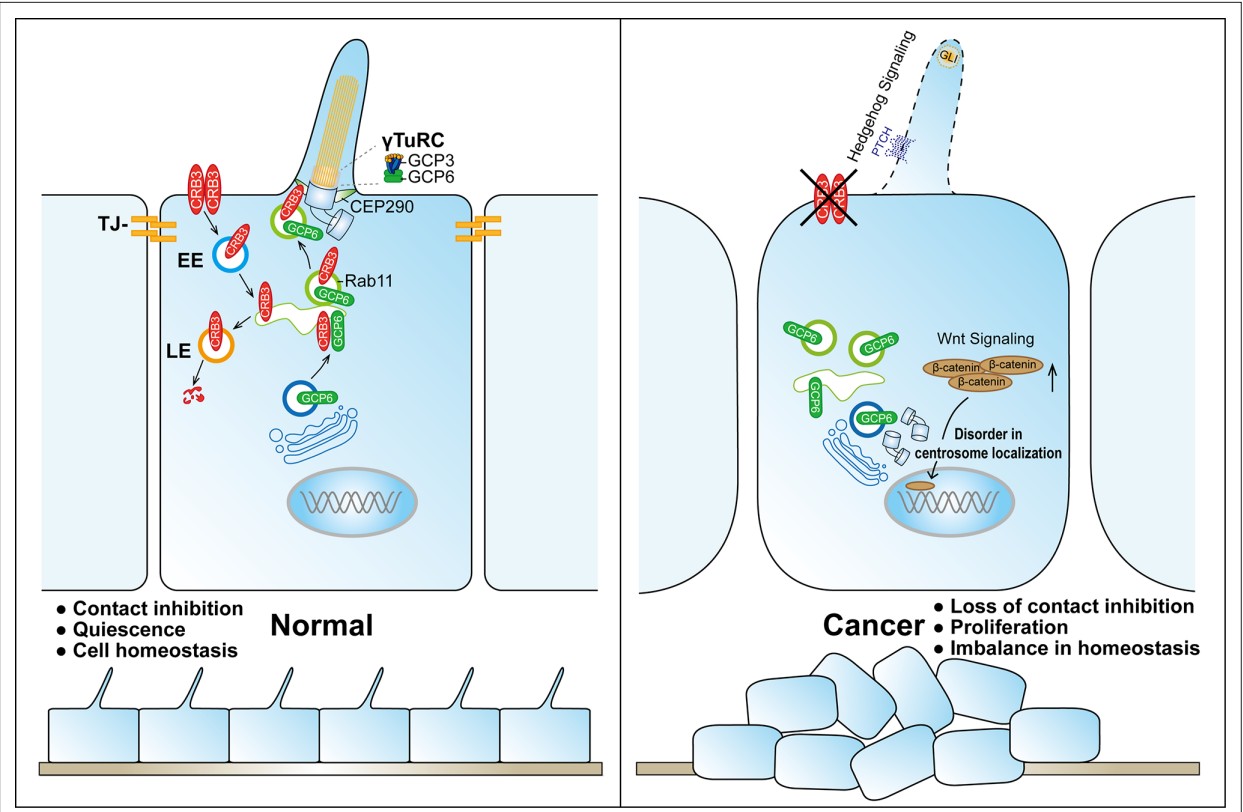

**Figure 9.** Schematic model of CRB3 regulating ciliary assembly. Graphic summary of prominent phenotypes observed after CRB3 deletion. CRB3 is localized on apical epithelial surfaces and participates in tight junction formation to maintain contact inhibition and cell homeostasis in quiescence. Inside the cell, Rab11-positive endosomes mediate the intracellular trafficking of CRB3, and CRB3 navigates GCP6/Rab11 trafficking vesicles to CEP290. Then, GCP6 is involved in normal γ-tubulin ring complex (γTuRC) assembly in ciliogenesis. In CRB3 deletion cells, the primary cilium fails to assemble properly, and the Wnt signaling pathway is activated through β-catenin upregulation and nuclear localization, but the Hh signaling pathway fails to be activated. This cellular imbalance is disrupted, leading to tumorigenesis. TJ, tight junction; EE, early endosome; LE, late endosome.

Here, we reveal that Rab11-positive endosomes mediate the intracellular trafficking of CRB3 and that CRB3 can navigate GCP6/Rab11 trafficking vesicles to CEP290, promoting correct γTuRC assembly during primary cilium formation. When cells form tight junctions, CRB3 is mainly localized to the cell membrane, and subsequently, the endosomes participate in the intracellular degradation process of CRB3 on the cell membrane. The intracellular CRB3 can bind to Rab11, which in turn participates in primary cilium assembly. Additionally, in the presence of CRB3, the epithelium can form contact inhibition and an intact primary cilium in quiescence to achieve cellular homeostasis. In contrast, the loss of contact inhibition and primary cilium formation with CRB3 deletion can activate the Wnt signaling pathway, affect the Hh signaling pathway, and disrupt the imbalance of this cellular homeostasis, resulting in significant cell proliferation during tumorigenesis (*Figure 9*).

The formation of the primary cilium has been divided into several distinct phases. It begins with the maturation of the mother centriole, also known as the centriole-to-basal-body transition (*Sánchez and Dynlacht, 2016*; *Kobayashi and Dynlacht, 2011*). DAVs, which are small cytoplasmic vesicles originating from the Golgi and the recycling endosome, accumulate in the vicinity of the distal appendages of the mother centriole (*Lu et al., 2015*; *Kobayashi et al., 2014*; *Schmidt et al., 2012*). In growing RPE1 cells, Rab8a is localized to cytoplasmic vesicles and the Golgi/trans-Golgi network, while Rab8-positive vesicles are rapidly redistributed to the mother centriole by binding to Rab11-positive recycling and endosomes under serum starvation (*Westlake et al., 2011*; *Das and Guo, 2011*). The EHD1 protein converts these DAVs to form larger ciliary vesicles, which elongate through continuous fusion with Rab8-positive vesicles to produce the primary cilium membrane (*Lu et al., 2015*; *Knödler et al., 2010*; *Rainey et al., 2010*). Therefore, Rab11-positive recycling endosomes, as important transporters, are necessary for primary cilium formation and centriole-basal-body transition in the early phase. Our

study found that CRB3 could bind to some members of Rab small GTPase family, significantly participating in Golgi vesicle transport and vesicle organization pathways through mass spectrometry identification. Coexpression and co-IP results showed that CRB3 interacted with Rab11, which mediated the intracellular trafficking of CRB3 within Rab11-positive endosomes at different cell densities.

Unlike motile cilia, primary cilia have a 9+0 structure, consisting of only nine outer doublet microtubules without a central pair. Similar to microtubule assembly at microtubule-organizing centers (MTOCs) during mitosis, γ-tubulin, which is a part of γTuRC, is also concentrated in the basal body of the cilium in interphase (*Guichard et al., 2023*). γ-Tubulin complexes have been shown to form microtubule templates and regulate microtubule nucleation through longitudinal contacts with α-tubulin and β-tubulin (*Kollman et al., 2011*). The γTuRC consists of multiple copies of γTuSC, which is composed of two copies of γ-tubulin and one each of GCP2 and GCP3. Then, GCP2 and GCP3 interact with GCP4, GCP5, and GCP6 to form γTuRC (*Liu et al., 2021*). Our experiments showed that CRB3 knockdown did not affect the expression of the γTuRC-specific proteins GCP3, GCP6, and γ-tubulin, but inhibited the interaction between GCP6 and GCP3, leading to the destabilization of γTuRC. Further analysis revealed that significant co-localization was observed between GCP6- and Rab11-positive vesicles at low cell density, while at high cell density, GCP6 accumulated at the centrosome with GCP3. CRB3 knockdown resulted in the disappearance of this accumulated GCP6. These results indicated that GCP6/Rab11 trafficking vesicles must be transported to the basal body for ciliogenesis. While it is still unclear how CRB3 interferes with γTuRC assembly during primary cilium formation, we hypothesize that Rab11-positive endosomes mediate the intracellular trafficking of CRB3 and that CRB3, as a navigator interacting with CEP290, can navigate GCP6/Rab11 trafficking vesicles to the transition zone for γTuRC assembly during primary cilium formation.

CEP290 is a centriolar or ciliary protein that localizes to the centrosomes in dividing cells, the distal mother centriole in quiescent cells, and the transition zone in the primary cilium (*Chen et al., 2021*). Mutations in *CEP290* are associated with various diseases, such as the devastating blinding disease Leber's congenital amaurosis, nephronophthisis, Senior Løken syndrome, Joubert syndrome, Bardet–Biedl syndrome, and lethal Meckel–Gruber syndrome (*Drivas and Bennett, 2014*). In particular, the phenotypes of CEP290-associated ciliopathies are very similar to those in our *Crb3* knockout mouse model, with lethality and ocular and renal abnormalities. CEP290 knockdown significantly decreased the number of cells with primary cilia, and CEP290 knockout mice showed mislocalization of ciliary proteins in the retinas (*Tsang et al., 2008*; *Chang et al., 2006*). Additionally, the depletion of CEP290 disrupts protein trafficking to centrosomes and affects the recruitment of the BBSome and vesicular trafficking in ciliary assembly (*Kim et al., 2008*; *Stowe et al., 2012*). Although CEP290 serves as a hub to recruit many ciliary proteins, it has not been reported to be involved in polarized vesicle trafficking or localization. Our study verified that CEP290 could interact with the polarity protein CRB3. CRB3 knockdown disrupts the interaction between CEP290 and Rab11-positive endosomes. Thus, these results validate our hypothesis. Although CRB3 navigates GCP6/Rab11 trafficking vesicles to CEP290, we further need to detect the region of CRB3 interacting with CEP290 and what other cargos are transported by CRB3/Rab11 trafficking vesicles in ciliary assembly.

In addition, our results showed that CRB3 knockdown failed to activate the Hh signaling pathway with SAG in MCF10A cells. Under untreated conditions, PTCH1 located at the ciliary membrane inhibits SMO, and Suppressor of Fused (SuFu) sequesters Gli transcription factors, leading to their degradation. SAG can activate the Hh signaling pathway, resulting in dissociation into the nucleus to regulate the expression of downstream target genes (*Wheway et al., 2018*). The Hh signaling pathway mainly regulates cellular growth and differentiation and is abnormal in different types of tumors. Some studies indicate that cilia are double-sided, promoting and preventing cancer development through the Hh signaling pathway in vivo. It is thought that cilia deletion could activate SMO to inhibit tumor growth, while promoting carcinogenesis was induced by activated Gli2 (*Wong et al., 2009*). Although SAG cannot activate the Hh signaling pathway in CRB3-depleted cells, further research is needed to determine the role of CRB3 in the carcinogenic process through the ciliary Hh signaling pathway in PyMT-cKO-Crb3 mice. Additionally, another canonical Wnt signaling pathway was altered after CRB3 knockdown. The Wnt signaling pathway is an important cascade regulating development and stemness, and aberrant Wnt signaling has been reported in many cancer entities, especially colorectal cancer (*Zhan et al., 2017*). In our study, CRB3 downregulated the expression of β-catenin. Thus, CRB3 can affect cilium-related signaling pathways in tumorigenesis.

In summary, our study provides novel evidence for the role of the apical polarity protein CRB3 in regulating viability, mammary and ocular development, and primary cilium formation in germline and conditional knockout mice. Deletion of CRB3 caused irregular lumen formation and ductal epithelial hyperplasia in mammary epithelial-specific knockout mice. Further study uncovered a new function of CRB3b in interacting with Rab11, navigating GCP6/Rab11 trafficking vesicles to CEP290 for γTuRC assembly during ciliogenesis. In addition, CRB3 also participates in regulating cilium-related Hh and Wnt signaling pathways in tumorigenesis. Understanding these polarity protein-mediated vesicle trafficking mechanisms will shed new light on luminal development and tumorigenesis.

## Materials and methods

### Generation of floxed *Crb3* mice

All animal experiments were verified and approved by the Committee of Institutional Animal Care and Use of Xi'an Jiaotong University (permit number: XJTUAE2018-1801). All surgeries were performed under sodium pentobarbital anesthesia, and every effort was made to minimize suffering. Heterozygous $Crb3^{wt/fl}$ mice (C57BL/6J) were generated using Cas9/CRISPR-mediated genome editing (Cyagen Biosciences, Guangzhou, China). The gRNA to *Crb3* and Cas9 mRNA was coinjected into fertilized mouse eggs to generate targeted conditional knockout offspring. The *Crb3* gRNA sequences were gRNA1, 5'-GGCTGGGTCCACACCTACGGAGG-3'; gRNA2, 5'-ACCCACAAAGCCACGCCA GTGGG-3'. Mouse genotyping was screened using PCR with the primers F1, 5'-ACATAAGGCCTTCCGTTAAG CTG-3'; R1, 5'- GTGGATTCGGACCAGTCTGA-3'. $Crb3^{wt/fl}$ mice were intercrossed with MMTV-Cre mice or CAG-Cre mice to generate tissue-specific *Crb3* knockout mice. MMTV-Cre mice (FVB), CAG-Cre mice (C57BL/6N), and MMTV-PyMT mice (FVB) were purchased from Cyagen Biosciences. All mice were bred in the specific pathogen-free (SPF) animal houses of the Laboratory Animal Center of Xi'an Jiaotong University.

Mice were genotyped using PCR and DNA gel electrophoresis. Genomic DNA from the mouse tail was extracted using the Mouse Direct PCR Kit (Bimake, TX, #B40015) according to the manufacturer's instructions. The genotyping primers were F3, 5'-TTGAGAGTCTTAAGCAGTCAGGG-3'; R5, 5'-AACC TTTCCCAGGAGTA TGTGAC-3'. PCR results showed that $Crb3^{wt/wt}$ was one band with 163 bp, $Crb3^{wt/fl}$ was two bands with 228 bp and 163 bp, and $Crb3^{fl/fl}$ was one band with 228 bp. The primers for identifying *Cre* alleles were F, 5'-TTGAGAGTCTTAAGCAGTCAGGG-3'; R, 5'-TTACCACTCCCAGCAAGACA C-3', and amplification with one 247 bp band. The reaction conditions of *Crb3* and *Cre* PCR were as follows: 94°C for 5 min, 94°C for 20 s, 60°C for 30 s, and 72°C for 20 s for 35 cycles.

### Mouse mammary gland whole-mount analysis

After euthanizing the mice, the inguinal mammary gland tissues were removed intact and fully spread onto slides. The tissues were rapidly fixed in Carnoy's fixative for 2 hr at room temperature. Subsequently, the tissues were washed in 70% ethanol solution for 15 min at room temperature, and in 50 and 30% ethanol and distilled water for 5 min each. Staining was performed in carmine alum solution at 4°C overnight. After washing with a 70% ethanol solution, the tissues were washed again with 70, 95, and 100% ethanol for 15 min each. Tissues were cleared in xylene overnight at room temperature and then mounted with neutral balsam. Images were obtained using a microscope (Leica DMi8, Wetzlar, Germany).

### Cell culture and transfection

The MCF10A, MCF7, T47D, MDA-MB-231, HCC1806, and MDA-MB-453 cell lines were purchased from Shanghai Institute of Biochemistry and Cell Biology (National Collection of Authenticated Cell Cultures, Shanghai, China). These cell lines have been authenticated by NCACC, and we have confirmed via PCR testing that they are negative for mycoplasma contamination. MCF10A cells were routinely grown in DMEM/F12 (1:1) media (HyClone, UT) supplemented with 5% horse serum, 20 ng/ml human EGF, 10 µg/ml insulin, and 0.5 µg/ml hydrocortisone. MCF7, MDA-MB-231, and MDA-MB-453 cells were cultured in high glucose DMEM media (HyClone) supplemented with 10% FBS (HyClone, #SH30084.03). T47D and HCC1806 cells were grown in RPMI-1640 media (HyClone) supplemented with 10% FBS. All cell lines were incubated in 5% $CO_2$ at 37°C.

siRNAs were purchased from GenePharma company (Shanghai, China). Cultured cells were transfected with siRNAs or plasmids using Lipofectamine 2000 (Invitrogen, CA, #11668-019) following the manufacturer's protocol. The negative control shRNA (NC) and shRNA against CRB3 (both isoforms CRB3a and CRB3b) were packaged into lentivirus as previously described (*Li et al., 2017*; *Mao et al., 2017*).

## DNA constructs and stable cell lines

CRB3b was PCR amplified from RNA extracted from MCF10A cells and cloned into the pCMV-Blank vector (Beyotime, Shanghai, China, D2602) using T4 ligase (NEB, MA). Then, the GFP sequence was inserted between the extracellular and transmembrane CRB3 domains to generate the pCMV-CRB3-GFP vector, as described previously (*Djuric et al., 2016*). The full-length CRB3 (1–120), CRB3 (1–116), CRB3 (1–83), CRB3 (1–58), and CRB3 (1–26) were amplified by PCR from the pCMV-CRB3-GFP vector using specific primers to create EcoR I and Xho I restriction sites, and then cloned into the pCMV-Blank vector. The full-length was also cloned into the lentiviral vector pLVX-TetOne-Puro (Clontech, Takara, #631849). The plasmid pECMV-3×FLAG-RAB11A was purchased from Sino-Biological (Beijing, China). The plasmids of pECMV-3×FLAG-RAB11A[S20V]/[S25N]/[Q70L] mutants were constructed using the *Fast* Mutagenesis System (Transgen, Beijing, China, FM111-01).

Lentiviruses were produced in HEK293T cells. CRB3-GFP in MCF10A and MCF7 cells were generated using a lentivirus expression system. Stable cell lines were screened using puromycin after lentiviral infection for 72 hr.

## 3D morphogenesis

To initiate 3D morphogenesis, growth factor-reduced Matrigel (Corning, #354230) was added to a four-well chamber slide system (Corning, #177437) and then incubated at 37°C for solidification. MCF10A cells were plated into this chamber slide and cultured in DMEM/F12 (1:1) media (HyClone) supplemented with 2% horse serum, 5 ng/ml human EGF, 10 μg/ml insulin, 0.5 μg/ml hydrocortisone, and 2.5% Matrigel. 3D morphogenesis was photographed using a microscope (Leica DMi8) at different time points. On day 14 after cell culture, 3D MCF10A cells were fixed and stained.

## Immunohistochemistry

The paraffin-embedded sections of samples were baked, deparaffinized in xylene, and sequentially rehydrated in gradient concentrations of ethanol, sequentially. Then, antigen retrieval was performed in Tris-EDTA buffer (pH 9.0) heated to 95°C for 20 min. Endogenous peroxidase activity was reduced in 3% hydrogen peroxide for 10 min at room temperature. Then, 5% goat plasma was used for blocking at 37°C for 30 min, and the sections were incubated with specific primary antibodies overnight at 4°C. Primary antibodies used were Ki67 (Abcam, UK, ab279653), phospho-histone H3 (Abcam, ab267372), cleaved caspase 3 (CST, USA, #9661), GLI1 (CST, #3538), and β-catenin (CST, #8480). The following steps used biotin-streptavidin HRP detection kits (ZSGB-BIO, China, SP-9001, SP-9002) according to the manufacturer's instructions. The sections were stained with DAB (ZSGB-BIO, ZLI-9017), counterstained with hematoxylin, dehydrated in gradient concentrations of ethanol, cleared in xylene, and mounted with neutral balsam, sequentially. Images were obtained using a slide scanner (Leica SCN400).

## Immunofluorescence

Cells were plated on coverslips and fixed in 4% paraformaldehyde for 30 min. Then, the membranes were permeabilized using 0.2% Triton X-100. The cells were blocked with 5% BSA solution for 1 hr at room temperature. Cells were incubated with specific primary antibodies overnight at 4°C. Caspase 3 (CST #9662S), α-tubulin (CST, #3873), acetylated tubulin (Proteintech, USA, #662001-IG), γ-tubulin (Proteintech, #15176-1-AP), pericentrin (Abcam, ab4448), CRB3 (Sigma, USA, HPA013835), EEA1 (BD, USA, #610456), Rab11 (BD, #610823), and GCP6 (Abcam, ab95172) were used as primary antibodies. The secondary antibodies were Alexa Fluor 488-labeled or 594-labeled (Invitrogen, A32731, A32744). DPIA (5 μg/ml) was used for DNA staining, and images were captured using a confocal microscope (Leica SP5 II).

## MEFs isolation and maintenance

The embryos were harvested from *Crb3*^wt/fl^;CAG-Cre mice cross at E13.5, and the head, tail, and internal organs were removed. The tail was used for genotyping, and DNA was extracted from it.

Then, the body was minced and digested with trypsin in a 37°C incubator for 30 min. The resulting cells were resuspended in MEF media (DME, 10% FBS, 1% penicillin/streptomycin) and plated onto 10 cm cell culture dishes with one embryo in each. They were subsequently passaged 1:3 with MEF media to expand the culture.

### Real-time PCR assay

Total RNA from cell lines was isolated using TRIzol reagent (Invitrogen, #15596026), and 5 μg RNA was converted to cDNA with the RevertAid first strand cDNA synthesis kit (Thermo, USA, K1622) according to the manufacturer's instructions. Real-time PCR was prepared with SYBR qPCR Premix (Takara, Japan, RR420L), and then performed with a real-time PCR detection instrument (Bio-Rad CFX96, USA). The primers used for real-time PCR were purchased from Tsingke Biotech (Beijing, China), and the sequences are listed in *Supplementary file 1*. The mRNA expression was normalized to *GAPDH*, and fold changes were calculated using the ΔΔCt method.

### Immunoprecipitation and immunoblotting

Cultured cells were lysed in RIPA buffer supplemented with protease inhibitors (Roche, NJ). Then, the cell lysates were subjected to SDS-PAGE separation and were transferred to PVDF membranes. The membranes were subjected to immunoblot assays using antibodies against CRB3 (Santa Cruz, sc-292449, both isoforms CRB3a and CRB3b can be detected), GCP2 (Santa Cruz, sc-377117), GCP3 (Santa Cruz, sc-373758), GCP4 (Santa Cruz, sc-271876), GCP5 (Santa Cruz, sc-365837), GCP6 (Abcam, ab95172), γ-tubulin (Proteintech, #66320-1-Ig), Rab11 (Abcam, ab128913), GFP (Roche, #11814460001), Flag (Sigma, F7425), CEP290 (Santa Cruz, sc-390637), GSK3-β (CST, #12456), β-catenin (CST, #8480), GAPDH (Proteintech, #HRP-6004), and β-actin (Proteintech, #HRP-60008). HRP-conjugated secondary antibodies (CST, #7074, #7076) were then incubated at room temperature for 1 hr in the dark. Final detection was achieved by using ECL Plus (Millipore, Germany, WBULS0500). Proteins for the immunoprecipitation assay were lysed with NP40 buffer. The cell lysates were detected by a Dynabeads protein G immunoprecipitation kit (Invitrogen, #10007D) following the manufacturer's instructions. The elution protein complexes were subjected to an immunoblot assay.

### Protein identification and bioinformatics analyses

LC-MS/MS analysis was conducted by PTM Bio (Zhejiang, China). Q Exactive Plus (Thermo, MA) was used for tandem mass spectrometry data analysis. The Blast2GO program against the UniProt database was used to perform functional annotation and classification of the identified proteins. Subsequently, DAVID tools were used to perform pathway enrichment analysis.

### Scanning electron microscope (SEM) observations

MCF10A cells with CRB3-GFP and CRB3 knockdown were plated on culture slides, and doxycycline (Dox) was added to induce the expression of CRB3-GFP. After the cells were fully confluent, the slides were fixed with 4% paraformaldehyde at 4°C overnight, followed by 1% osmium tetroxide at 4°C for 1 hr. Dehydration was performed at room temperature, and the samples were dried using the critical point drying method. Images were obtained by using a SEM (Hitachi TM1000, Japan).

### Clinical data

Patient data, breast cancer tissues, and adjacent para-cancerous tissues were collected from the First Affiliated Hospital of Xi'an Jiaotong University (Shaanxi, China). All patients signed informed consent forms before surgery, and their characteristics are listed in *Supplementary file 2*. This research was authorized by the Ethics Committee of the First Affiliated Hospital of Xi'an Jiaotong University (permit number: 2018G-127) and conducted in conformity with the Declaration of Helsinki.

### Statistics

Statistical analysis was performed using SPSS statistics 23.0 for Windows (IBM, Armonk). All experiments were repeated at least three times. The values are expressed as the mean ± standard deviation (SD). Unpaired Student's $t$-test was used to compare the differences between two groups. One-way ANOVA followed by Dunnett's multiple-comparisons test was used for multiple comparisons. The $\chi^2$

test was used to assess the significance of the observed frequencies. $p < 0.05$ is considered an indicator of statistical significance.

## Acknowledgements

This work was supported by grants from the National Natural Science Foundation of China (no. 81872272 and 82173023), Innovation Capability Support Program of Shaanxi (no. 2020TD-046), and Clinical Research Award of the First Affiliated Hospital of Xi'an Jiaotong University (no. XJTU1AF-CRF-2017-007). We are grateful to Prof. Ceshi Chen (Kunming Institute of Zoology, China) and Prof. Yongping Shao (Xi'an Jiaotong University, China) for their guidance in this research. We also thank Qi Tian, Lizhe Zhu, and Yan Zhou for their help with the experiments.

## Additional information

### Funding

| Funder | Grant reference number | Author |
|---|---|---|
| National Natural Science Foundation of China | 81872272 | Peijun Liu |
| National Natural Science Foundation of China | 82173023 | Peijun Liu |
| Innovation Capability Support Program of Shaanxi | 2020TD-046 | Peijun Liu |
| Clinical Research Award of the First Affiliated Hospital of Xi'an Jiaotong University | XJTU1AF-CRF-2017-007 | Peijun Liu |

The funders had no role in study design, data collection and interpretation, or the decision to submit the work for publication.

### Author contributions

Bo Wang, Validation, Investigation, Writing – original draft, Writing – review and editing; Zheyong Liang, Validation, Investigation, Writing – original draft; Tan Tan, Yina Jiang, Yangyang Shang, Validation, Investigation; Miao Zhang, Xiaoqian Gao, Shaoran Song, Juan Li, Data curation; Ruiqi Wang, He Chen, Jie Liu, Investigation; Yu Ren, Writing – original draft; Peijun Liu, Conceptualization, Supervision, Funding acquisition, Validation, Investigation, Writing – original draft, Writing – review and editing

### Author ORCIDs

Bo Wang https://orcid.org/0000-0001-7633-4435
Peijun Liu http://orcid.org/0000-0003-0529-387X

### Ethics

Human subjects: Patient data, breast cancer tissues, and adjacent para-cancerous tissues were collected from the First Affiliated Hospital of Xi'an Jiaotong University (Shaanxi, China). All patients signed informed consent forms before surgery. This research was authorized by the Ethics Committee of the First Affiliated Hospital of Xi'an Jiaotong University (Permit Number: 2018-G-127) and conducted in conformity with the Declaration of Helsinki.

All animal experiments were verified and approved by the Committee of Institutional Animal Care and Use of Xi'an Jiaotong University (Permit Number: XJTUAE2018-1801). All surgery was performed under sodium pentobarbital anesthesia, and every effort was made to minimize suffering.

Reviewer #1 (Public Review): https://doi.org/10.7554/eLife.86689.4.sa1
Author Response https://doi.org/10.7554/eLife.86689.4.sa2

## Additional files

### Supplementary files
- Supplementary file 1. The sequences of primer pairs used in real-time PCR.
- Supplementary file 2. Characteristics of patients.
- MDAR checklist

### Data availability
All data generated or analysed during this study are included in the manuscript and supporting file; Source Data files have been provided for Figure 5. Figure 5 - Source Data 1 contains the numerical data used to generate the figure.

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
