## [Editor Report · eLife assessment]

This is a **useful** study for scientists interested in cell polarity, epithelial morphogenesis, cancer, and primary cilia. The authors investigate the role of CRB3 in regulating these processes by using a combination of a mammary epithelial cell-specific conditional Crb3 knockout mouse model, and cellular, molecular and biochemical approaches. The results, which are **solid**, supporting and extending previous findings, suggest that CRB3 affects ciliogenesis by a mechanism involving Rab11 and gamma-TuRC.

---

## [Referee Report · Reviewer #1 (Public Review)]

In this study the authors first perform global knockout of the gene coding for the polarity protein Crumbs 3 (CRB3) in the mouse and show that this leads to perinatal lethality and anopthalmia. Next, they create a conditional knockout mouse specifically lacking CRB3 in mammary gland epithelial cells and show that this leads to ductal epithelial hyperplasia, impaired branching morphogenesis and tumorigenesis. To study the mechanism by which CRB3 affects mammary epithelial development and morphogenesis the authors turn to MCF10A cells and find that CRB3 shRNA-mediated knockdown in these cells impairs their ability to form properly polarized acini in 3D cultures. Furthermore, they find that MCF10A cells lacking CRB3 display reduced primary ciliation frequency compared to control cells, which is supported by rescue experiments and is in agreement with previous studies implicating CRB3 in primary cilia biogenesis. Using a combination of biochemical, molecular- and imaging approaches the authors then provide evidence indicating that CRB3 promotes ciliogenesis by mediating Rab11-dependent recruitment of gamma-tubulin ring complex (gamma-TuRC) component GCP6 to the centrosome/ciliary base, and they also show that CRB3 itself is localized to the base of primary cilia. Finally, to assess the functional consequences of CRB3 loss on ciliary signaling function, the authors analyze the effect of CRB3 loss on Hedgehog and Wnt signaling using cell-based assays or a mouse model.

Overall, the described findings are interesting and in agreement with previous studies showing an involvement of CRB3 in epithelial cell biology, tumorigenesis and ciliogenesis. The results showing a role for CRB3 in mammary epithelial development and morphogenesis in vivo seem convincing. Although the authors provide evidence that CRB3 promotes ciliogenesis via (indirect) physical association with Rab11 and gamma-TuRC, the precise mechanism by which CRB3 promotes ciliogenesis remains to be clarified.

---

## [Author Response]

The following is the authors’ response to the previous reviews

**Reviewer # 1 (Public Review)**
Specific comments1. For all cell-based assays using shRNA to knock down CRB3, it would be desirable to perform rescue experiments to ensure that the observed phenotype of CRB3 depleted cells is specific and not due to off-target effects of the shRNA.

Thank you for your comments. Based on your suggestions, we performed the rescue experiments to observe any alterations in the primary cilia of CRB3-depleted MCF10A cells with overexpressed CRB3. The revised parts can be found in lines 186-188 and the new Figure 3-Supplement Figure 1A-C has been added.

1. Figure 3G: it is very difficult to see that the red stained structures are primary cilia.

Yes, the staining structure of primary cilia in mammary ductal lumen are less clear than that of individual cells and in renal tubule in Figure 3G. We used recognized acetylated tubulin and γ-tubulin to stain the primary cilia, which were clearly labeled in individual cells. However, the labeled primary cilia in renal tubule were longer length and demonstrated a more pronounced structure than those in the mammary ductal lumen. In the mammary ductal lumen of the 10 mice we analyzed, the primary cilia showed shorter length and staining structure than the others shown in Figure 3G. This difference may be due to the distinct characteristics of primary cilia in different tissues.

1. Figure 5A: it is unfortunate the authors chose not to show the original dataset (Excel file) used for generating this figure; this makes it difficult to interpret the data. It is general policy of the journal to make source data accessible to the scientific community.

In accordance with the journal policy, we have provided the original dataset (Excel file) for Figure 5A, as detailed in “Figure 5–Source Data 1”.

1. The authors have a tendency to overinterpret their data, and not all claims put forth by the authors are fully supported by the data provided.

We have carefully read through the whole text and have revised the overinterpretation parts. These parts can be found in lines 48-50, lines 93-95, and lines 260-261.

**Reviewer # 2 (Public Review)**

Thank you for recognizing and supporting our research for this manuscript.

**Reviewer # 1 (Recommendations For The Authors)**
1. Abstract line 48-51: data overinterpretation. The authors cannot claim this based on the data they are presenting. Please modify the statement/temper the claims.

Thanks for your comments. We have revised this sentence in the abstract, as well as lines 48-50 for details.

1. There are several grammatical errors throughout the manuscript. In particular, the following sentences/statements are either wrong, confusing or non-sensical: lines 55-56; lines 87-90; lines 93-95; lines 385-387; lines 409-410.

Thanks for your positive comments. We have modified lines 55-56 to become new lines 54-55. These sentences in lines 87-90 and lines 93-95 are difficult to understand and logically problematic, so we have carefully revised this paragraph (new lines 85-90). Lines 385-387 have been deleted as they are non-sensical. Lines 409-410 contain misrepresentations. We have revised them in new lines 408-409.

1. Lines 257-259: this is data over-interpretation. It is not correct to state CRB3 is highly dynamic without having done any live cell imaging.

Thank you for your comments. We have revised this sentence, see revised lines 260-261 for details.

1. Figure 8E: if cells do not make cilia when CRB3 is lost (Figure 3), how is it possible to analyze SMO localization to cilia in these cells?

Thank you for your comments. We used immunofluorescence techniques, with acetylated tubulin and SMO co-staining, to analyze the localization of SMO to cilia. The results of immunofluorescent staining of primary cilium and statistical analysis in Figure 3 showed that the proportion of cells with primary cilium was significantly lower in the CRB3 knockdown group, but cells with primary cilium were still present. We used laser confocal microscopy micrographs to identify cells with primary cilium by staining acetylated tubulin, then analyzed the co-localization under the SMO channel, and finally analyzed the proportion of SMO-positive cilia. Several publications (J Cell Biol. 2020;219(6):e201904107; Science. 2008;320(5884):1777-81; Proc Natl Acad Sci U S A. 2012;109(34):13644-9.) have demonstrated that knocking down genes can affect primary cilium formation, and this method has also been used to examine the localization of SMO-related signaling pathway molecules on primary cilium.

1. Lines 366-366: based on the relative low magnification of the images in Figure 8H it is difficult to assess the subcellular localization of GLI1 and whether there is a difference between wild type and the Crb3 mutant cells. For example, it is not clear if GLI1 is localizing to the centrosome-cilium axis. Please modify the text accordingly.

Thank you for your good suggestions. As you mentioned, IHC cannot observe the subcellular localization of GLI1 on the centrosome-cilium axis. However, since GLI1 is a transcriptional effector at the terminal end of the Hh signaling pathway, we may not have made it clear that what we observed in the IHC results was the localization of GLI1 in the nucleus. Therefore, we have revised the description accordingly, as described in line 368 and lines 520-521.

1. Figure 7D, E: the zoomed-in images look pixelated.

Thank you for your positive comments. We have replaced these images in the new Figure 7D and E.

1. Figure 8B: Acetylacte-tub is misspelled.

Thank you for your comments. We have revised and standardized the acetylated tubulin stain to "Ace-tubulin" in all immunofluorescent images throughout the manuscript.

**Reviewer # 2 (Recommendations For The Authors)**
CRB3 is present in mammals as 2 isoforms, A and B, originating from an alternative splicing. In this study, the authors never mention this fact and when using approaches to KO or KD CRB3A/B they are likely to deplete both isoforms which have been shown to have different C-terminal domains and functions (Fan et al., 2007). This is also important for the CRB3 antibodies used in the study since according to the material and methods section they are either against the extracellular domain common to both isoforms or the intracellular domain which is only similar in the domain close to transmembrane between the 2 isoforms. Since the antibodies used in each figure are not detailed it is impossible to know if the authors are detecting CRB3A or B or both. Please provide the information and correct for the actual isoform detected in the data and conclusions.From the revised version we know now that CRB3B is used for exogenous expression. It has been shown that each isoform has a different role and localization in cells so why focus only on CRB3B for this study?

Thank you for your positive comments. First, previous literature has reported that CRB3b localizes in the primary cilium of MDCK cells. We have corrected the Introduction to specify CRB3b (line 81). Secondly, in the methodology section, we show that the CDs sequence of CRB3b was PCR-amplified from RNA extracted from MCF10A cells. We also designed primers specific to CRB3a but were unable to amplify them, indicating that CRB3b is significantly more expressed in epithelial cells than CRB3a. Finally, according to the company recommended by Genecards website for purchasing CRB3 cloning products, the only CRB3 sequence available in the CRB3 cDNA ORF Clone in Cloning Vector, Human (Cat: HG14324-G) from Sino Biological is CRB3b.

The authors use GFP-CRB3A/B, it is not stated which isoform, over-expression to localize CRB3A/B in MCF10A cells (figure 4A). The levels of expression appear to be very high in the GFP panel and it is likely that the secretory pathway of the cells is clogged with GFP-CRB3A/B in transit from the ER to the plasma membrane. Thus, the co-localization with pericentrin might be due to the accumulation of ER and Golgi around the centrosome. This co-localization should be done with the endogenous CRB3A/B and with a better resolution.The authors do not answer about the potential mislocalization of overexpressed exogenous protein.

We acknowledge the reviewer's perspective. The large amount of exogenous protein overexpression in the cell could potentially obstruct the protein secretion pathway, resulting in the accumulation of the exogenous protein at the ER and Golgi. Such accumulation could create the false impression of co-localization between CRB3b and the centrosome. To provide additional details (lines 215-217 and lines 426-433), we re-expressed the results exogenously and subsequently used staining of endogenous CRB3 and γ-tubulin in Figure 4C to confirm the co-localization of CRB3 and the centrosome.

The staining for CRB3A/B in Figure 4C (red) is striking with a very strong accumulation in an undefined intracellular structure and the authors do not provide any explanation for such a difference with the GFP-CRB3A/B just above.The authors explain that two different photonic techniques are used (classical versus confocal) but in a cell biology manuscript confocal microscopy is now the standard technique.

Thank you for your comments. We have included a discussion on the partial concordance between CRB3's endogenous staining and exogenous expression results in the "Discussion" section, specifically in lines 420-435.

In addition, the authors claim (Line 251/252) that Rab11 is necessary for the transport of CRB3A/B but they should KD Rab11 to show this.The author's answer is that blocking endocytosis with dynasore is as good as knocking down Rab11 to show its interaction and role in CRB3A/B transport which is not the case.

Thank you for your comments. As requested by the reviewers, we have conducted experiments to knockdown Rab11 and detect CRB3 intracellular trafficking, as shown in the new Figure 5-Supplement 2B and added lines 258-260. These results provide additional support for our conclusions.

The domain of CRB3A/B that is necessary for the interaction with Rab11 is the N-terminal part of the extracellular domain. This domain is thus inside the transport vesicles and not accessible from the cytoplasm. Given that Rab11 is a cytoplasmic protein, how the 2 proteins could interact across the membrane? The authors do not even discuss this essential point for their hypothesis.Comment on the revised version: the authors still do not understand the basic of cell biology since they claim that the extracellular domain of CRB3 can be in contact with Rab11 after endocytosis. Even after endocytosis the extracellular domain of CRB3A/B is inside the lumen of the endosome and not in contact with the cytosol where Rab11 is located. Lines 420-421 of the revised manuscript still claim this interaction between the two proteins without providing the link between the cytosol where Rab11 is and the endosome lumen where the extracellular domain of CRB3A/B is. Please correct.

Thank you for your positive comments. After carefully studying the relevant knowledge, we strongly agree with the reviewer's point of view. We have toned down our claim and removed the description regarding the binding of Rab11 endosomes to specific structural domains of intracellular CRB3 that we were unable to confirm (see lines 443-444 and lines 465-466).